# Exploring the Limitations of Layer Synchronization in Spiking Neural Networks

**Roel Koopman**                                                    *roel.koopman@cwi.nl*
*Machine Learning Group, Centrum Wiskunde & Informatica (CWI)*
*Amsterdam, The Netherlands*

**Amirreza Yousefzadeh**                                            *a.yousefzadeh@utwente.nl*
*Faculty of Electrical Engineering, Mathematics & Computer Science, University of Twente*
*Enschede, The Netherlands*

**Mahyar Shahsavari**                                               *mahyar.shahsavari@donders.ru.nl*
*Donders Centre for Cognition, Radboud University*
*Nijmegen, The Netherlands*

**Guangzhi Tang**                                                   *guangzhi.tang@maastrichtuniversity.nl*
*Department of Advanced Computing Sciences, Maastricht University*
*Maastricht, The Netherlands*

**Manolis Sifalakis**                                               *manolis.sifalakis@innatera.com*
*Innatera Nanosystems BV*
*Rijswijk, The Netherlands*

**Reviewed on OpenReview:** *https://openreview.net/forum?id=mfmAVwtMIk*

## Abstract

Neural-network processing in machine learning applications relies on layer synchronization. This is practiced even in artificial Spiking Neural Networks (SNNs), which are touted as consistent with neurobiology, in spite of processing in the brain being in fact asynchronous. A truly asynchronous system however would allow all neurons to evaluate concurrently their threshold and emit spikes upon receiving any presynaptic current. Omitting layer synchronization is potentially beneficial, for latency and energy efficiency, but asynchronous execution of models previously trained with layer synchronization may entail a mismatch in network dynamics and performance. We present and quantify this problem, and show that models trained with layer synchronization either perform poorly in absence of the synchronization, or fail to benefit from any energy and latency reduction, when such a mechanism is in place. We then explore a potential solution direction, based on a generalization of backpropagation-based training that integrates knowledge about an asynchronous execution scheduling strategy, for learning models suitable for asynchronous processing. We experiment with 2 asynchronous neuron execution scheduling strategies in datasets that encode spatial and temporal information, and we show the potential of asynchronous processing to use less spikes (up to 50%), complete inference faster (up to 2x), and achieve competitive or even better accuracy (up to ~10% higher). Our exploration affirms that asynchronous event-based AI processing can be indeed more efficient, but we need to rethink how we train our SNN models to benefit from it. (Source code available at: `https://github.com/RoelMK/asynctorch`)

## 1 Introduction

Artificial Neural Networks (ANNs) are the foundation behind many of the recently successful developments in AI, such as in computer vision (Szegedy et al., 2017; Voulodimos et al., 2018) and natural language processing

(Vaswani et al., 2017; Brown et al., 2020). To match the complexity of the ever more demanding tasks, networks have grown in size, with advanced large language models having billions of parameters (Zhao et al., 2024). With this, the power consumption has exploded (Luccioni et al., 2023), limiting the deployment to large data centers. In an effort to learn from our brain's superior power efficiency, and motivated by neuroscience research, SNNs (Maass, 1997) bolster as an alternative. They use discrete binary or graded spikes (events) for communication, are suited for processing sparse features (He et al., 2020), and when combined with event-based asynchronous processing are assumed to reduce latency and improve energy efficiency(Zeki, 2015). Sparsity can save synaptic operations, thus lowering energy consumption, and asynchronous operation enables concurrent evaluation of all neurons anywhere in the network purely event-driven, leading to low latency.

In an *event-based* processing system (figure 1.A), a spike or event in an SNN (or non-zero activation in an ANN) can independently and locally trigger some downstream computations (that may or may not have a global effect). Event-based processing manifests essentially by comparison to vector-based processing, which groups (activation) computations in batches.

By *asynchronous processing* we understand the fact that spikes (or activations) can be processed at any time (or in any order), and also anywhere in the network, independently of any other spikes and unhindered by artificial conditions that delay their immediate propagation (such as waiting for all other neurons in the same layer to integrate their currents or evaluate their thresholds). Typically, for a large network of neurons only the approximate timing of events plays a role in asynchrony (London et al., 2010), and the resulting *order* is in often the main information-carrying signal (according to neuroscience literature that underpins order codings) (Thorpe et al., 2001). Approximate or relative timing makes asynchronous processing viable in digital accelerators even though they discretize time. Asynchronous processing, as perceived in this work, is illustrated (and visually represented through an analogy) in figure 1.C

It needs to be meticulously clarified that asynchronous processing within the network, and at the hardware processing system level (which is the focus of this work), is distinct from what is customarily referred to as asynchrony at the model input. Specifically, at the model level in RNNs, we colloquially refer to *timesteps* as the discretization of continuous external stimuli into a sequence of consecutive steps for input to the network model. Asynchrony in the latter sense (as commonly seen in SNN literature) refers to the fact that the stimulus is sampled across the input dimension and can be presented to the model not in a single step, but rather distributed i.i.d. across several such discrete timesteps. To limit ambiguity, we call this

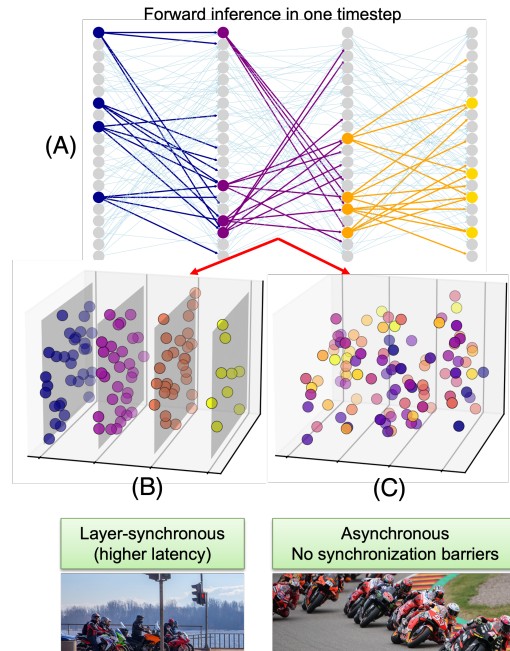

Figure 1: (A) Event-based processing. (B) Event-based processing with per-layer synchronization: latency is a function of the events in the system as well as the number of layers (synchronization barriers). (C) Asynchronous (end-to-end) event-based processing: event processing does not encounter synchronization barriers, and so inference latency is determined solely by the subset of the total number of events processed until a decision is made at the output layer. Color codes indicate at which layer events were fired. The x-axis of the plots represents time/order that events were processed. The vertical planes in (B) represent layer-synchronization barriers (also indicating bottlenecks in network routing and memory I/O). The difference between (B) and (C) can be understood through an analogy: bikes racing in a circuit versus on a public road with a sequence of traffic lights.

*input asynchrony* [1], and we distinguish it from (in-)*network asynchrony* which we study in this paper, namely the relative propagation of spatial signals *inside* the network within individual timesteps. [2] Moreover, with input asynchrony, by common convention, processing of the stimulus in one timestep inside the network has been assumed until now to be as taking place "atomically" through a deterministic process and sequence: breadth-first across neurons one-layer-after-the-other. This convention serves the purpose that spikes (activation state) from two distinct timesteps, but also between layers, do not interfere with one another. Forward inference in two timesteps interacts only through "neuron state caching" in each layer, by means of (not resetting) the membrane voltages (hidden state in RNNs). But, from the viewpoint of this work, this type of processing inside the network is still synchronous (despite input asynchrony) as depicted in figure 1.B. Moreover, we argue this per-layer synchronization dulls the dynamics of the stimulus (data-generating process); is not quite what happens in the brain (SNNs); and neither is the type of processing neuromorphic processors are primarily aimed for.

Asynchronous processing in the brain (Zeki, 2015) has been connected to energy and latency efficiency because information dynamics can express freely and propagate signals promptly through neuron clusters triggering early outputs and leveraging fast decisions even before other neuron clusters of slower dynamics have finished processing the input stimulus (and generated downstream activations). Specifically, there is no known process in the brain that consistently and uniformly forces neurons in one layer (or column) to finish their computations before the neurons in another downstream layer assume processing of their input currents. On the contrary, the more intensely the stimulus manifests in some regions (subgroups of neurons), the faster dynamics are reinforced, leading to a non-uniform (unaligned) and non-synchronized propagation of spikes across regions. These dynamics are also likely modulating and/or are modulated along certain pathways by heterogeneity in axonal or synaptic delays and different state decay processes across neurons.

In other words, through asynchronous processing, in-network expressed dynamics should lead to computation economy. The work in this paper puts this hypothesis to test. It presents a first of its kind study of how current synchronous training impacts such an asynchronous modus operandi, how can we possibly train good performing robust models for asynchronous processing, and if energy and latency efficiency stem from such a modus operandi.

Using a simulation environment that implements the concept of network asynchrony and can emulate the processing behavior of a number of neuromorphic accelerators, we provide quantitative results on benchmark datasets (with different spatio-temporal information content) and network topologies of two or more hidden layers, which show performance degradation and latency/energy inefficiency resulting from changes in model dynamics when trained with layer synchronization and later deployed for asynchronous inference (processing). Next, we explore a potentially promising solution by proposing a generalization of gradient (backpropagation-based) training, that can be parameterized with various neuron execution scheduling strategies for asynchronous processing, and vectorization abilities present in various neuromorphic processors. We show that using this training method, it is possible not only to recover the compromised accuracy, but also to fulfil the expectation of saving energy and improving latency under asynchronous processing (when compared to the conventional breadth-first processing in GPUs). This work opens a path for design space explorations aimed to bridge the efficiency gap between neural network model training and asynchronous processor design.

## 2 Related work

Conventional highly parallel ANN compute accelerators, such as Graphics Processing Units (GPUs) and Tensor Processing Units (TPUs), which are primarily optimized for dense vectorized matrix operations, face inherent challenges in exploiting unstructured and temporal sparsity for improving their energy efficiency.

---

[1] Notably a few approaches, such as in Zhu et al. (2022) quote as asynchrony the manipulation through learning of the shape and distribution of stimulus (input spikes) across timesteps with the goal of improving model performance (accuracy), assuming however execution with layer-synchronized processing (event-based or vectorized alike).

[2] As will be seen later, we also need a notion of discretization of (processing) steps through the network, which can be variable and decoupled from the number of layers. We refer to these steps as *forward steps*. Thus, while timesteps discretize processing (in complete forward model evaluations) across stimulus presentations, *forward steps* discretize processing within a timestep from the onset of (one timestep's) stimulus presentation until the output neurons conclude the model evalution for that timestep.

Targeting the common practice of executing ANNs with per-layer synchronization bottlenecks has left them with poor support for asynchronous processing (for improving latency or energy consumption). At best, they parallelize processing within a layer and/or pipeline processing across layers. This leaves an exploration space for neuromorphic processors that try to excel in handling the event-driven nature of SNNs and leverage asynchronous concurrent processing, offering efficiency advantages in various tasks.

However, the training of SNNs today more and more often relies conveniently, for performance, on conventional end-to-end ANN training methods (Dampfhoffer et al., 2023), based on synchronizing computations per-layer rather than event-driven per neuron (Guo et al., 2023a). First, the total of all presynaptic currents from the preceding layer must be computed and integrated before postsynaptic neurons in the current layer update their state and evaluate their activation function (i.e. emit new spikes). For consistency, neuron evaluation in one layer must thus complete and synchronize before proceeding to evaluate neurons of the next layer.

Some event-driven neuromorphic processors, such as μBrain (Stuijt et al., 2021), Speck (Caccavella et al., 2023), and Pulsar's (Inn, 2024) SNN sub-system, are fully event-driven and lack any layer synchronization mechanism. They are capable of fully asynchronous processing but this makes them notoriously difficult to train out-of-the-box with methods for ANNs that rely on per-layer synchronization (e.g. (Dampfhoffer et al., 2023)). Speck developers propose to train their models with hardware in-the-loop (Liu et al., 2024) to reduce the mismatch between the training algorithms and the inference hardware. However, this method does not provide a general solution for training asynchronous neural networks. Others, like SpiNNaker (Furber et al., 2014) and TrueNorth (Akopyan et al., 2015), use timer-based synchronization, or Loihi (Davies et al., 2018) and Seneca (Tang et al., 2023) use barrier synchronization between layers, in order to ensure consistency and alignment with models trained in software (with layer synchronization). As we will show, this entails an efficiency penalty.

Functionally, the most suitable type of model for end-to-end asynchronous processing is probably rate-coded SNN models, whereby neurons can integrate state and communicate independently from any other neuron. These models are trained like ANNs (Zenke & Vogels, 2021; Lin et al., 2018; Diehl et al., 2015) or converted from pre-trained ANNs (Rueckauer et al., 2017; Kim et al., 2020). Therefore, to be accurate under asynchronous processing, it is required to run the inference for a long time, reducing the latency and energy benefit of using SNNs (Sengupta et al., 2019).

Alternative to rate-coding models are temporal-coding models, with time-to-first spike (TTFS) (Park et al., 2020; Srivatsa et al., 2020; Kheradpisheh & Masquelier, 2020; Comşa et al., 2022) or order encoding (Bonilla et al., 2022). They are very sparse (hence energy efficient) but very cumbersome and elaborate to convert from ANNs (Painkras et al., 2013; Srivatsa et al., 2020) or train directly (Kheradpisheh & Masquelier, 2020; Comşa et al., 2022), less tolerant to noise (Comşa et al., 2022), and their execution so far, while event-based, requires some form of synchronization or a reference time. Efficiency is thus only attributable to the reduced number of spikes, all of which need to be evaluated in order before a decision is reached. Other alternative encodings for event-based processing of SNNs include phase-coding (Kim et al., 2018) and burst coding (Park et al., 2019), which are, however, no more economical than rate coding and have not been shown, to our knowledge, to attain competitive performance.

The approach presented in this paper is, in fact, unique in enabling the trainability of models for event-based asynchronous execution, providing efficiency from processing only a subset of spikes, and delivering consistent performance and tolerance to noise. Also relevant to the works in this paper are SparseProp (Engelken, 2024) and EventProp (Wunderlich & Pehle, 2021b) on efficient event-based simulation and training, respectively. EventProp (Wunderlich & Pehle, 2021b) is potentially more economical than discrete-time backpropagation for training event-based models for asynchronous processing (and fully compatible with the work in this paper), but it has not been shown how its complexity scales beyond 1-2 hidden layers. SparseProp (Engelken, 2024) proposes an efficient neuron execution scheduling strategy for asynchronous processing in software simulation. An effective hardware implementation of this scheduling strategy can be found in (Monti et al., 2017), which has inspired the herein proposed *"momentum schedule"*. Moreover, we advance this research by demonstrating how to train SNN models using this event scheduler.

the fact that neurons evaluate their threshold immediately upon receiving a synaptic input (spike) from each fan-in synapse independently (as opposed to aggregating all synaptic inputs and the evaluating the threshold

once – where they can often cancel each other out); the relaxation of forcing synchronization of processing at every layer; and the adoption of dynamic scheduling of neuron execution across the entire network which continuously adapts (stochastically) to flow dynamics of stimulus/activations or neuron membrane states. The last last two "ingredients" allow neurons deep inside the network to fire before processing at early layers has finished (if the flow dynamics encourage it).

## 3    Methods for simulating and training of asynchronous SNNs

In this section, we detail our methodology. First, we detail (section 3.1) how we simulate the unconstrained behavior of various (digital) neuromorphic event-based accelerators capable of asynchronous SNN processing inside the neural network (i.e., implement *network asynchrony*), by leveraging the following principles: (a) allow neurons to evaluate their thresholds independently for every received synaptic current; (b) abolish the constraint to synchronize activations at each layer; (c) adopt dynamic scheduling of neuron execution across the entire network that continuously adapts to the flow dynamics of activations/stimulus or relative neuron states; and (d) support limited-length vectorization. (Details about the accelerators we considered, and the parameterizability of the simulator to support them, are given in the tables in appendix A.5)

Then in section 3.2 we explain how these principles can be applied to backpropagation training (leading to an approach we call *unlayered backpropagation*) to prepare good performing, robust and efficient asynchronous models.

### 3.1    Simulating asynchronous SNNs

The SNNs used in this work consist of $L$ layers of Leaky Integrate-and-Fire (LIF) neurons (He et al., 2020), with each layer $l$ for $1 \leq l \leq L$ having $N^{(l)}$ neurons and being fully-connected to the next layer $l + 1$, except for the output layer. Each input feature is connected to all neurons in the first layer. Every connection has a synaptic weight. Using these weights, we can compute the incoming current $x$ per neuron resulting from spikes, as per equation 3 in appendix A.1.

Each LIF neuron has a membrane potential exponentially decaying over time based on some membrane time constant $\tau_m$. We use the analytical solution (see equation 4 in appendix A.1) to compute the decay. By keeping track of the membrane potential $u[t]$ and the elapsed time $\Delta t$ since time $t$, it is possible to precisely calculate the membrane potential for $t + \Delta t$. Therefore, computations are required only when $x[t + \Delta t] > 0$.

To determine if a neuron spikes, a threshold function $\Theta$ checks if the membrane potential exceeds a threshold $U_{\text{thr}}$, following equation 5 in appendix A.1. If $\Theta(u) = 1$, the membrane potential is hard reset to 0. Soft reset is another option (Guo et al., 2022) as well as refractoriness (Sanaullah et al., 2023). However, for simplicity here we only experiment with hard reset.

#### 3.1.1    Asynchronous inference with event-driven state updates

We can vectorize the computations introduced in the previous section on a per-layer basis. This is herein referred to as the "layered" inference approach, which implies layer synchronization, since all neurons of one layer need to evaluate their state, before any neuron in a down-stream layer does the same. An alternative to this situation is an event-driven approach, where the computations for state updates are applied in response to new spike arrivals. If additionally spikes can be emitted (threshold evaluation) at any point in time (e.g in response to any individual synaptic current), by any neuron in the network, affecting the states of postsynaptic neurons independently of others, then (we assume) this approach can achieve a true representation of network asynchrony.

When using network asynchrony, the network dynamics evolve with each spike. Any single spike can generate multiple currents downstream, linked to a specific stimuli at timestep $t$, determined by the initiating input activity. Let $K_t$ represent the atomic computation that is executed at one neuron in the network, in response to input synaptic currents and triggered by stimulus presented at the network at timestep t, and which will update the state of the neuron and evaluate its threshold function (activation). The exact computations for a LIF neuron can be found in section A.2.

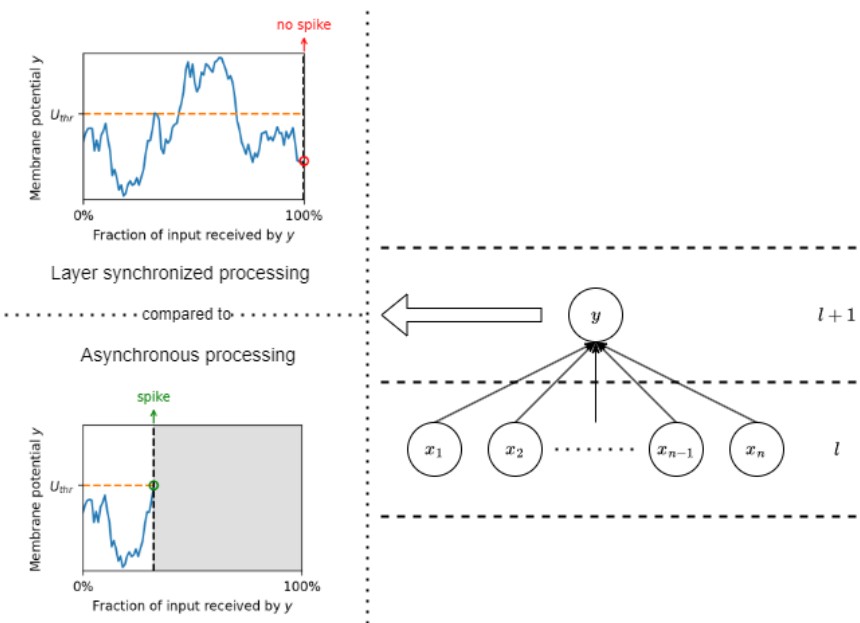

Figure 2: Layer synchronization ensures that activations from all neurons ($x_i$) in a layer are available simultaneously to any post-synaptic neuron ($y$) in the next layer. Hence they are integrated together and the threshold of the post-synaptic neuron is evaluated only once (see top-left graph). As a result the currents arriving from different synapses often cancel each other out given opposite sign weights, resulting in fewer spikes and a different firing pattern than when each synaptic current is processed independently (see bottom-left graph), where each input is more likely to trigger a spike, as is the case in asynchronous processing. Thus, in SNNs, layer synchronization leads to the volume of spikes being reduced and the activation flow dynamics being dulled. This is not an issue (at least not as grave) in ANNs because activations communicate relative magnitudes.

Notice that whether $K_t$ executes in response to input current from each synapse independently, or in response to the aggregate synaptic current from all fan-in synapses collectively, leads to different activation behavior/patterns and hence neuron dynamics (see figure 2). Also, the relative order in which $K_t$ executes among neurons across the entire network (as opposed to within just one layer – due to synchronization) affects the output of the network due to the non-linearity entailed in $K_t$. Here, for simplicity, it is assumed that the fan-out currents resulting from a spike generation are propagated as a group at once (e.g. no heterogeneous propagation delays exist). Because of network asynchrony however the recipient neurons will *not necessarily* process them simultaneously or in fixed order, but rather opportunistically as dictated by a (dynamic) scheduling algorithm. This dynamic scheduling captures the asynchronous hardware behaviour/capabilities (more about this is given in section 3.1.2). Because of these assumptions, analyzing the spike propagation order gives the same insights as analyzing the execution order of $K_t$ (across neurons in the network).

Overall, unlike the layered approach, in this *modus-operandi* because of the per-input current evaluation of the firing threshold neurons can fire more than once at the same timestep $t$ (in response to the same input stimulus). We can prevent this by inducing a neuron to enter a refractory state for the remainder of time $t$. In this state, the neuron keeps its membrane potential at 0. This makes the model causally simpler, more tractable, and more "economic" when energy consumption is coupled to spike communication. These aspects may also give explicit retrospective justification for the neurophysically observed refractoriness (Berry & Meister, 1997). However, the observations we report in the results hold true both when we allowed neurons in the network to enter refractoriness after firing a spike, as well as when there is no refractoriness.

**Two resolutions of time** We apply input framing as typically done for inputs from a DVS sensor, accumulating timestamped events in discrete time-bins, and providing these frames as inputs to the network

---

**Algorithm 1** Vectorized network asynchrony forward pass

---

**Input:** input spikes $\mathbf{s}_{\text{in}} \in \mathbb{N}^{N_{\text{in}}}$, previous forward pass time $t_0 \in \mathbb{R}$, current forward pass time $t_1 \in \mathbb{R}$, neuron state $\mathbf{u} \in \mathbb{R}^N$ at time $t_0$, forward group size $F \in \mathbb{N}_{>0}$
**Output:** spike count $\mathbf{c} \in \mathbb{N}^N$
    $\Delta t \leftarrow t_1 - t_0$
    $\mathbf{u} \leftarrow \text{NeuronDecay}(\mathbf{u}, \Delta t)$
    $\mathbf{x} \leftarrow \text{InputLayerForward}(\mathbf{s}_{\text{in}})$
    $\mathbf{s} \leftarrow \mathbf{0}^N$                                                           ▷ Vector with zeros of length $N$
    $\mathbf{c} \leftarrow \mathbf{0}^N$
    **while** $(\text{Sum}(\mathbf{x}) \neq 0 \textbf{ or } \text{Sum}(\mathbf{s}) \neq 0) \textbf{ and } \neg\text{EarlyStop}(\mathbf{s})$ **do**
        $\mathbf{s}_{\text{new}}, \mathbf{u} \leftarrow \text{NeuronForward}(\mathbf{x}, \mathbf{u})$
        $\mathbf{s} \leftarrow \mathbf{s} + \mathbf{s}_{\text{new}}$                                                 ▷ Enqueue new spikes
        $\mathbf{c} \leftarrow \mathbf{c} + \mathbf{s}_{\text{new}}$                                               ▷ Update spike count
        $\mathbf{s}_{\text{selected}} \leftarrow \text{SelectSpikes}(\mathbf{s}, F)$
        $\mathbf{s} \leftarrow \mathbf{s} - \mathbf{s}_{\text{selected}}$                                        ▷ Dequeue selected spikes
        $\mathbf{x} \leftarrow \text{NetworkLayersForward}(\mathbf{s}_{\text{selected}})$
    **end while**

---

in discrete *timesteps*. The *timestep size* refers to the time-interval between the timestamps of the first (or last) events in the time-bins of two consecutive timesteps. Timesteps correspond to one concept of timing (resolution) with same semantics as in RNNs, and which we refer to here as *macro-time*. At each timestep, input event frames initiate network activity, in a so-called *forward pass*, making all new spikes contingent on the timestep.

The order of spikes propagated though the network during a timestep establishes another notion of "time" in a forward pass (we call it "micro-time"), which is discretized in so-called *forward steps*. A forward step is associated with the processing of one or a group of events (in case of vectorization), and the number of forward steps measures time to complete inference. One can realize that in case of layered inference the number of forward steps is fixed and equals the number of layers, but this is not the case for asynchronous processing.

### 3.1.2 Vectorized network asynchrony

The event-driven (neuron state) update rules for network asynchrony as introduced in the previous section can be vectorized by selecting a number of spikes for processing them simultaneously. This allows us to consider the entire spectrum of possibilities between per-layer synchronization at one extreme (by assuming a vector size equal to a layer size), and "complete" asynchrony at the other extreme, where each spike event is processed entirely independently of all others. Additionally, vectorization makes acceleration possible by exploiting the parallelization features and vector pipelines of accelerators, where these models execute, leading to pragmatic simulation of network asynchrony.

During the simulation, the states of all the $N = \sum_{l=1}^{L} N^{(l)}$ neurons in the network are stored in vectors. Vector $\mathbf{x} \in \mathbb{R}^N$ tracks the computed input currents for the neurons, $\mathbf{u} \in \mathbb{R}^N$ the membrane potential of the neurons, $\mathbf{s} \in \mathbb{N}^N$ the emitted spikes awaiting processing, and $\mathbf{c} \in \mathbb{N}^N$ which neurons have spiked in the current forward pass. For any of those vectors, the indices from $\sum_{k=1}^{l-1} N^{(k)}$ to $\sum_{k=1}^{l} N^{(k)}$ represent the values for the neurons in layer $l$, for $1 \leq l \leq L$ where $N^{(0)} = 0$.

Algorithm 1 outlines the processing during a forward pass (propagation of the spikes of an input frame through the network). Input spikes are available in $\mathbf{s}_{\text{in}} \in \mathbb{N}^{N_{\text{in}}}$ (not to be confused with $\mathbf{s}$) where $N_{\text{in}}$ is the number of input features. Each *forward pass* consists of *forward steps* (the code within the while loop), which update the state of a set of neurons based on the spikes selected for propagation. A complete list of parameters can be found in section A.5.

The forward pass ends either when all spike activations have been processed ("On spiking done" stop condition), or when any neuron in the output layer has spiked one or more (default is one) forward steps ago ("On output" stop condition). The latter is checked in the EarlyStop function.

The SelectSpikes function defines how to select a subset of the emitted spikes for propagation. The function selects $F$ spikes at a time, or less if there are less than $F$ remaining spikes ready to be propagated. The method of spike selection is determined by a *scheduling policy*.

For the experiments here two policies are exercised. The intent is that different scheduling strategies can be tested for their effectiveness in capturing tractably asynchronous processing dynamics. The first, Random Scheduling (RS), randomly picks spikes from the entire network. The second, Momentum Scheduling (MS), prioritizes spikes from neurons based on their membrane potential upon exceeding their threshold.

The neuron model-specific (LIF) behavior is expressed in the NeuronDecay and NeuronForward functions. This entails the computations for state updates (see section 3.1) for all neurons in the network (NeuronDecay) or for only those neurons receiving input currents in the forward step (NeuronForward), with added restriction that spiking is only allowed once per forward pass.

The network architecture is defined by the InputLayerForward and NetworkLayersForward functions. These functions compute the values of synaptic currents from spikes. This follows from equation 3.

**Imitating neuromorphic accelerator hardware**   Neuromorphic processors come in a number of variations, but most of them have a template architecture that interconnects many tiny processing cores with each other. This design enables a scalable architecture that supports fully distributed memory and compute systems. In this template, each processing core has its own small synchronous execution domain, but the cores operate asynchronously among each other. Architectures such as those cited in the Section 2 all follow this template. Our asynchronous processing simulation environment captures the intrinsics of the synchronous domain (e.g. in the forward group size $F$) while simulating the asynchronous interactions among them (e.g. scheduling policy) and can be also configured to reproduce more specialized intrinsic behaviors of many of those processors (table 5).

## 3.2   Training asynchronous SNNs

We use backpropagation to train the model weights, and specifically when stimulous is presented across multiple timesteps (forward passes), such as for sequential or temporal data, then this is Backpropagation Through Time (BPTT). Following common practice to address the non-differentiability of the threshold function, the surrogate gradient method is used (Zenke & Vogels, 2021). Specifically the arctan function (Fang et al., 2021), (see section A.3 for details) provides a continuous and smooth approximation of the threshold function.

Class prediction is based on the softmaxed membrane potentials (for CIFAR-10) or spike counts over time (for the other datasets) of the neurons in the output layer, as described in section A.4. The loss is minimized using the Adam optimizer, with $\beta_1 = 0.9$ and $\beta_2 = 0.999$.

### 3.2.1   Unlayered backpropagation

We refer to "conventional" SNN training with per-layer synchronization using backpropagation (Dampfhoffer et al., 2023), as "layered backpropagation".

The vectorized network asynchronous processing approach is differentiable as well, and can be used with backpropagation. We refer to this as "unlayered backpropagation". Combined with BPTT unrolling, this method implies a two level unrolling. At the outer level, unrolling is based on discretization of time and thus the input across (as usually fixed number of) timesteps. At the inner level, unrolling is based on the $F$-grouping (and vectorized processing) of spikes in forward steps, subject to the scheduling policy, applied to the emitted spikes by neurons; from the beginning of the current timestep until the output is read.

Note, that the dependence on the scheduling policy, applied on a variable number of activations (across timesteps and data samples), in $F$-sized groups is the fundamental difference from layered backpropagation, and leads to different gradient state being built up in the computation graph. This state now captures the dynamics of asynchronous processing. For a single *backward pass* in a timestep of BPTT, which is applied in

a similar way as the layered backpropagation equivalent, it is given that:

$$\frac{\partial L_t}{\partial W} = \frac{\partial L_t}{\partial c_t} \sum_{i=1}^{N_t} \left( \frac{\partial c_t}{\partial s_i} \frac{\partial s_i}{\partial W} \right) \tag{1}$$

where $t$ is the time(step) of the *forward pass*, $W$ refers to the trainable weights, $L_t$ is the loss, $c_t$ is the spike count at the end of the forward pass, and $s_i$ is the emitted spikes vector at the end of the forward step $i$. The overall gradient (state) depends on the total number of *forward steps* $N_t$ in the forward pass and the spikes processed in each forward step. The number of steps scales linearly with the number of spikes processed in the forward pass, and it is important to understand that due to asynchronous processing the number of spikes processed until the evaluation of the loss function may be (well) less than the total number of spikes emitted throughout the forward pass. Since for every forward step, the computations are repeated, the time complexity scales linearly with the number of forward steps, $O(N_t)$. The same applies to the space complexity.

During the backward pass, the spikes in $s_i$ that are not selected for processing can skip the computation $f$ for that step:

$$\frac{\partial s_i}{\partial W} = \frac{\partial f(s_{i-1;\text{selected}}, u_{i-1}, x_t, W)}{\partial W} + \frac{\partial s_{i-1;\text{not selected}}}{\partial W} \tag{2}$$

Skip connections have been explored in deep ANNs and identified as a contributor to the stability of the training process (Orhan & Pitkow, 2018). This may apply to the skip in unlayered backpropagation, as well. The extent to which this is the case remains unexplored in this work.

Note that under this generalization, layered backpropagation corresponds to $F$-group the size of a layer, populated with a static round-Robin execution schedule following neuron index order within a layer, and re-reset across consecutive layers).

Finally, to leverage some visual insight into what makes unlayered backpropagation different from canonical (layered) backpropagation as a consequence of the ramifications we have proposed, figure 3 illustrates the forward and backward paths in the two cases (the forward path in figure 3.c depicts essentially the control-flow of our simulator). The most notable difference is that because of the way activation flow is generated in a variable number of forward steps (which, unless $F$-group has equal size to the number of neurons in a layer, is often more than the number of layers in the network), the unrolled compute graph in the backward pass will be different at every timestep, which contrasts with the fixed lattice-like structure of the compute graph in canonical backpropagation. This results in (fundamentally) different gradient flow and learning a different model (more robust for asynchronous inference, as our experiments will show).

### 3.2.2 Regularization techniques

During training, we use regularization to prevent overfitting and/or enhance model generalization. These techniques are not used during inference.

**Input spike dropout**. Randomly omits input spikes with a given probability. The decision to drop each spike is independent according to a Bernoulli distribution.

**Weight regularization**. Adds weight decay to the loss function: $L_{\lambda_W}(\mathbf{W}) = L(\mathbf{W}) + \lambda_W \|\mathbf{W}\|_2^2$ where $\lambda_W$ is the regularization coefficient, $L$ is the loss given the weights, and $\mathbf{W}$ are all the weights.

**Refractory dropout**. With some probability, do not apply the refractory effect, allowing a neuron to fire again within the same forward pass.

**Momentum noise**. When using momentum scheduling, noise sampled from $U(0,1)$ and multiplied by some constant $\lambda_{\text{MS}}$ is added to the recorded membrane potential while selecting spikes.

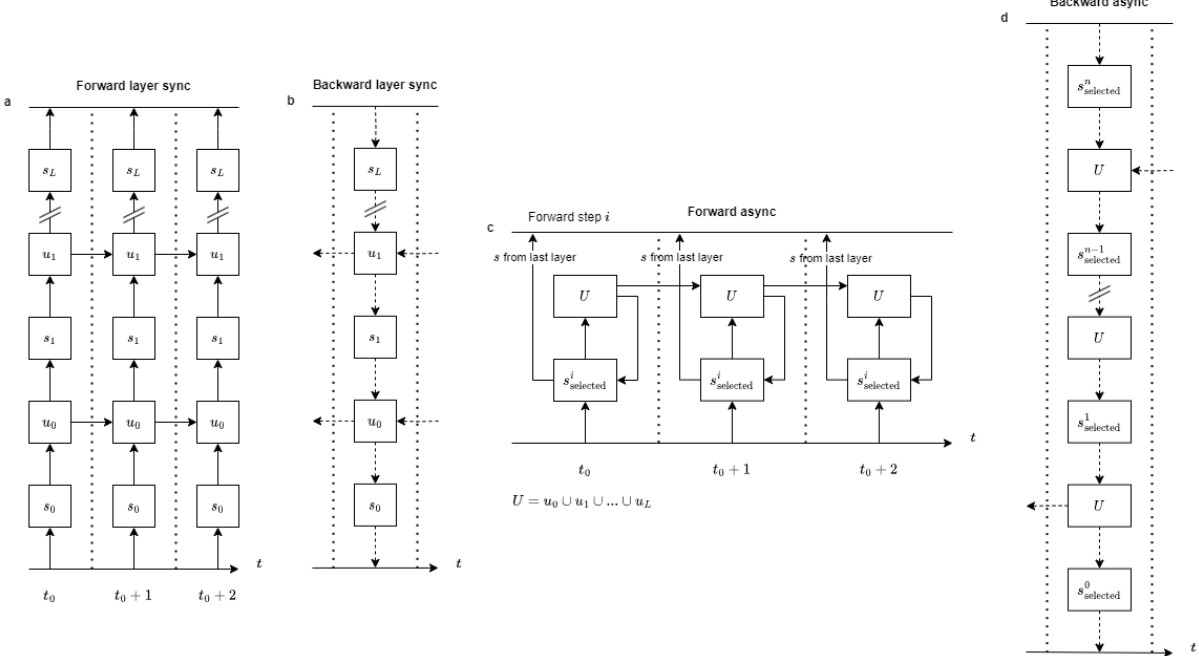

Figure 3: (a) The forward pass with layer synchronization, where layer $0 \leq j < L$ receives spike input $s_j$, creating currents that are integrated into membrane potentials $u_j$, which are carried over to the next timestep, and potentially emitting new spikes to the next layer if exceeding the spike threshold. (b) The backward pass of a single timestep for layered backpropagation. The structure of the computational graph is fixed across timesteps for a network with constant depth. (c) The forward pass for unlayered backpropagation, where $s^i_{\text{selected}}$ represents a selection of spikes from anywhere in the network propagated during the $i$th forward step, and $U$ denotes the membrane potentials of all neurons in the network. The membrane potentials are carried over to the next timestep at the end of the forward pass. Spikes from the last layer are pulled out of the loop as output. (d) The backward pass for unlayered backpropagation of a single timestep that consisted of $n + 1$ forwards steps. Unlike layered backpropagation, the structure of the computational graph varies across timesteps.

## 4 Results

### 4.1 Experimental setup

We carried out our experiments on SNN models trained primarily in three common benchmarking datasets, each of them has a different structure: N-MNIST (Orchard et al., 2015), SHD (Cramer et al., 2020), and DVS gestures (Amir et al., 2017). N-MNIST has purely spatial structure, SHD purely temporal, and DVS gestures combines both spatial and temporal (input framing in DVS gesture is done such that an entire gesture motion and contour is not revealed within a single frame). These three datasets were processed with fully connected network topologies, without explicit lateral connectivity. The input was provided sequentially across a number of timesteps, allowing the networks to operate recurrently, exploiting their statefulness (implicit recurrence) and neuron state decay, both of which are needed for temporal tasks. We also repeated some of the experiments with a fourth dataset, CIFAR-10 (Krizhevsky, 2012), in a more complex and deep convolutional VGG-style network. In this case, as the input consists of static images, it was fed to the network in a single timestep. Our experiments involved primarily models with rate coding without and with refractoriness forcing neurons down to few/single spikes per timestep. We did not experiment with population coding. More details on the datasets and network topologies can be found in sections A.6.1,A.10. Table 1 summarizes the parameterization of the experiments and the reported results. The network architecture and hyperparameters are given in table 6. State-of-the art performance for these tasks can be achieved with

Table 1: Parameterization of experiments and results.

| Parameterization | Description |
| --- | --- |
| Training method | Training with layered backpropagation is marked as "Layered" and with unlayered backpropagation as "Unlayered [scheduling strategy]". |
| Inference method | Inference with layer synchronization is marked as "Layered" and with network asynchrony as "Async [scheduling strategy]". |
| Scheduling strategy | How to select spikes from the queue. Can be random ("RS"), or based on the membrane potential just before spiking ("MS"). |
| Forward group size $F$ | Number of spikes to select for processing at the same time. Default is 8, both during training and inference. |
| Stop condition | Forward pass terminates: If all network activity drains ("On spiking done") or one forward step after the first spike is emitted by the output layer ("On output"). The default is "On spiking done" during training and "On output" during inference. |

reasonably shallow and wide models. We chose however to train narrow, but deeper network architectures so that the effects of absence of layer synchronization can be revealed in the comparison (because of this bias in network topology the accuracy results shown can vary from the top state of the art). In section A.7, we also provide an additional ablation with regard to how various training hyperparameters affect accuracy (forward group size, refractory dropout, and momentum noise during training). Results on error convergence are provided in section A.9.

## 4.2  Network asynchrony increases neuron reactivity

As observed in figure 4 (top row) during asynchronous inference, neurons are more reactive, i.e. a neuron can spike after integrating only a small number of incoming currents. With layer synchronization, this effect is averaged out as neurons are always required to consider all presynaptic currents (which are more likely to cancel each other out). For inference with $F = 8$, artifacts in the number of currents to spike can be observed, because each forward step propagates currents resulting from 8 spikes, causing neurons to integrate more currents than necessary for firing. Similar artifacts might also occur in neuromorphic chips equipped with fixed-width vector processing pipelines; making these observations insightful into the behavior of such hardware.

One expects that more reactive neurons imply higher activation density. Interestingly, this is not necessarily the case for the models trained for asynchronous inference! In figure 4 (bottom row) we see that if we wait for the network to "drain" of spike activity during a forward pass the total number of spikes will indeed be higher. But if the forward pass terminates as soon as a decision is made, asynchronous models are consistently sparser. This is because asynchronous processing allows spike activity to freely flow through to the output and not be blocked at every layer for synchronization.

## 4.3  Unlayered backpropagation recovers accuracy and increases sparsity

Network asynchrony negatively affects the performance of the models trained with layered backpropagation in all three datasets (table 2). This is likely the "Achilles' heel" of neuromorphic processing today. However, we observe that the accuracy loss is remediated when training takes into account asynchronous processing dynamics, which also significantly increases sparsity (by about 2x). We witnessed that the accuracy of models trained and executed asynchronously is consistently superior under the two scheduling policies we considered. This result conjectures that neuromorphic AI is competitive and more computationally efficient.

Figure 5 reveals another interesting result. It depicts how accuracy evolves as we allow more forward steps in the forward pass after the initial output during asynchronous inference. We see that because of the free flow of key information *depth-first*, models trained with unlayered backpropagation obtain the correct predictions as soon as the output layer gets stimulated. Activity that is likely triggered by "noise" in the input is integrated later on. Momentum scheduling is particularly good at exploiting this to boost accuracy.

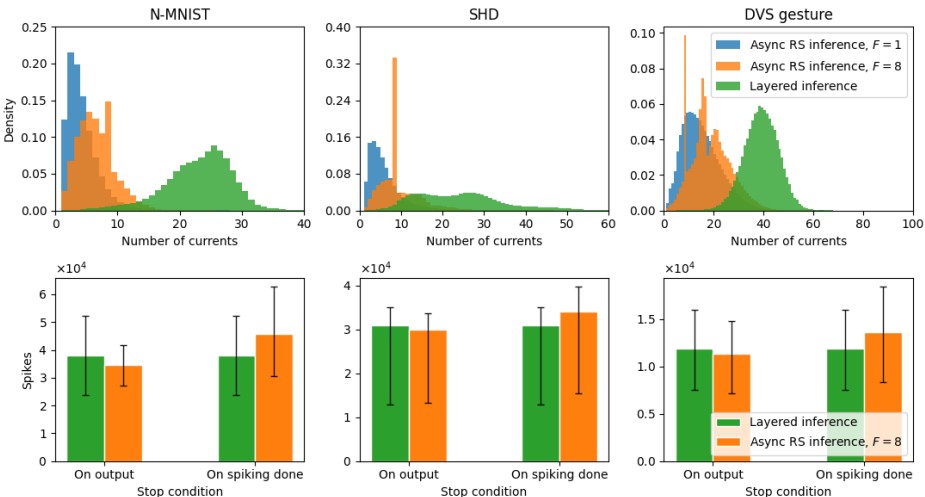

Figure 4: **(Top row)** Number of currents integrated by a neuron before spiking, recorded per neuron and per forward pass for all samples and neurons (excluding the neurons in the input layer). The Y-axis shows the relative frequency of the number of currents integrated before spiking. **(Bottom row)** Mean number of spikes per neuron during inference of all samples in the test. Error bars show the 25th and 75th percentiles. (Models in this figure were trained with layered backpropagation.)

Table 2: Accuracy and activation density results. More details about these metrics in section A.6.2.

| | | N-MNIST | | SHD | | DVS gesture | |
| Training | Inference | Acc. | Density | Acc. | Density | Acc. | Density |
|---|---|---|---|---|---|---|---|
| Layered | Layered | 0.949 | 3.987 | 0.783 | 14.386 | 0.739 | 46.473 |
| Layered | Async RS | 0.625 | 3.652 | 0.750 | 13.905 | 0.701 | 44.140 |
| Unlayered RS | Async RS | 0.956 | 1.504 | 0.796 | **5.100** | 0.777 | **25.140** |
| Unlayered MS | Async MS | **0.963** | **1.476** | **0.816** | 6.224 | **0.856** | 26.686 |

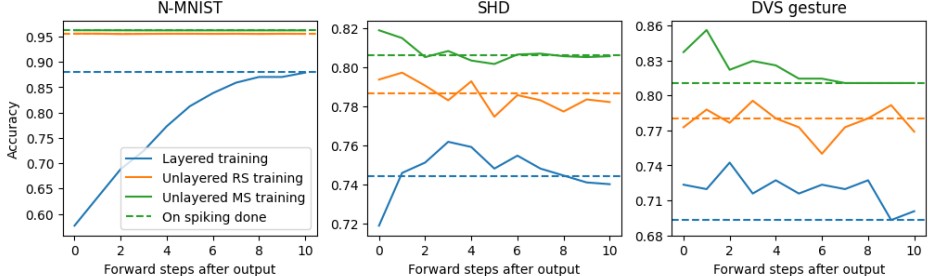

Figure 5: Accuracy as function of forward steps after the first spike in the output layer. Given that $F = 8$, each extra forward step processes another 8 spikes, assuming enough spikes are available. Dashed lines show the accuracy after all spike activity has been "drained" out of the network.

### 4.4 Network asynchrony and energy efficiency

To quantify the energy savings from asynchronous processing in the trained models, table 3 presents the mean number of synaptic operations and projected energy consumption on the µBrain neuromorphic chip (Stuijt et al., 2021) per sample. The reported reduction under asynchronous trained models with unlayered backpropagation account for savings in dynamic power. Additional details also on static power savings in a digital (neuromorphic) accelerator are provided in appendix A.11.

Table 3: Mean num of Synaptic Operations (SOs $\times 10^4$) and energy consumption (µJ) for classification of one sample, using 26 pJ/SO as measured for µBrain (Stuijt et al., 2021); ignoring static power. (Numbers are based on projections based on energy/SO reported in literature, not from actual on-hardware measurement.)

| Training | Inference | N-MNIST | | SHD | | DVS gesture | |
|---|---|---|---|---|---|---|---|
| | | SOs | Energy | SOs | Energy | SOs | Energy |
| Layered | Layered | 3.521 | 0.9155 | 50.82 | 13.21 | 158.8 | 41.29 |
| Layered | Async RS | 3.225 | 0.8386 | 49.12 | 12.77 | 150.9 | 39.22 |
| Unlayered RS | Async RS | 1.328 | 0.3454 | 18.02 | 4.684 | 85.92 | 22.34 |
| Unlayered MS | Async MS | 1.304 | 0.3389 | 21.99 | 5.717 | 91.2 | 23.71 |

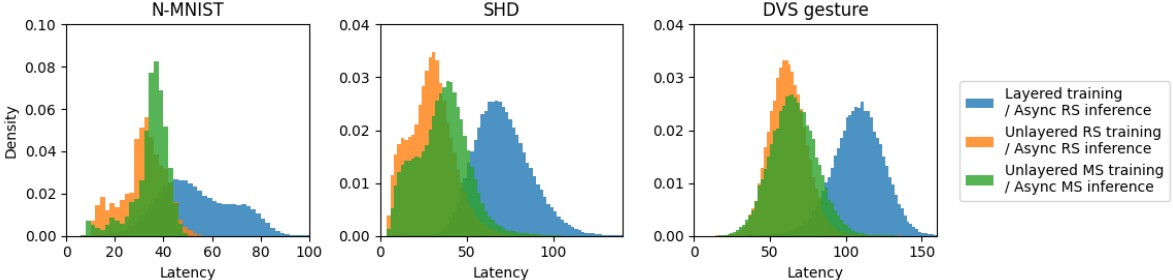

Figure 6: Latency per forward pass for all samples, in number of spikes until a decision in the output layer is reached. The Y-axis shows the relative frequency of recorded latencies.

### 4.5 Network asynchrony reduces latency

An equally important result concerns the inference latency reduction under asynchronous network processing. The models trained with unlayered backpropagation have significantly lower latency than those trained with layer synchronization. Assuming a unit latency for processing one spike, the comparatively worst case latency will be given under sequential processing, as a function of the number of spikes processed until an inference decision is made. Figure 6 shows a distribution of the inference latencies across the entire test-set. It should be clear by now that this is because "the important spikes" in these models quickly reach the output layer, uninhibited by layer synchronization.

### 4.6 Unlayered backpropagation is (currently) resource-intensive

We also tried to confirm the results in a scaled up setup, namely with a deeper VGG-7 like network (details in A.10), trained on the CIFAR-10 dataset. Figure 7a confirms the problem when training with layered backpropagation and the running asynchronous inference. When we try to train asynchronous models with unlayered backpropagation and the simpler RS scheduling policy the memory cost however becomes prohibitive unless we substantially increase the forward group size $F$. Nevertheless, even with as large as $F = 512$ (during training, the smallest possible with an NVIDIA RTX 5000 GPU) and tested with $F = 64$, accuracy steeply recovers to 71%, while activation density drops to about 1/4 (confirming our previous observations). We anticipate that with much smaller $F$ during training (higher degree of network asynchrony), the accuracy can be completely recovered.

Unfortunately, the computational cost of unlayered backpropagation, in the current framework (and implementation) is rather high, especially when training with a small $F$. Each time $F$ is halved, the time and memory requirements approximately double as discussed in sections 3.2.1 and A.8. While this computational cost imposes obvious scalability constraints in the current implementation of unlayered-backpropagation for training very deep models with complex structures, it does not burden however inference on-chip. But it creates a trade-off, since the costlier computationally the training becomes, the more robust asynchronous inference will be when the resulting model is deployed on-chip.

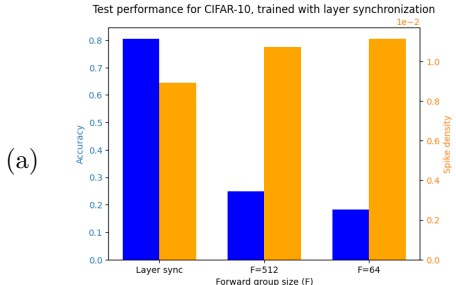

(a)                (b)

| Training | Inference | CIFAR-10 | |
| --- | --- | --- | --- |
| | | Acc. | Density |
| Layered | Layered | 0.805 | 0.0089 |
| Layered | Async RS | 0.183 | 0.0111 |
| Async RS | Async RS | 0.712 | 0.0026 |

Figure 7: (a) Impact of network asynchrony on a CIFAR-10 trained VGG-style model. A larger decrease in accuracy occurs with smaller forward group sizes accompanied by an increase in activations. (b) Accuracy and activation density results on CIFAR-10.

## 5 Discussion & Future outlook

We pinpoint a crux for neuromorphic AI processing in training SNN models conventionally with backpropagation and naively assuming they will execute consistently, and energy and latency efficiently in neuromorphic processors. The problem lies in neglecting the dynamics of asynchronous processing, and we found a way to factor them in the gradient training process. Resulting models not only recover the affected performance (and even exceed it) but also exhibit the energy and latency benefits touted by neuromorphic computing.

Asynchronous processing is mainly about neurons acting independently of other neurons (also across layer boundaries) based on locally emerging flow dynamics, which is feasible with rate-like codings. So long as these dynamics are unobstructed by synchronization barriers, there is no need for explicit time-tracking. Temporal coding schemes requiring explicit time tracking (timestamps) typically do so due to the presence of synchronization points in event processing. Moreover, order coding and TTFS schemes which may use timestamps to enforce strict per/across-layer ordering, can also do without them by working with queues. The overarching assumption is that exact timing is relevant mainly to the extent that it facilitates the relative ordering of events.

From this viewpoint, in our framework synaptic currents arriving at a neuron trigger *instantaneous evaluation of the membrane threshold* independently of other incoming currents (i.e. there is no integration time interval at the neuron level). This eliminates the need to keep an explicit global clock for spike times. The instantaneous membrane voltage of neurons then reflects the temporal dynamics of the incoming activations at the neuron level. The use of *dynamic scheduling* allows neuron order dynamics to be reflected in spike exchanges without synchronization delays at layer boundaries. In other words, these modeling ramifications reflect relative temporal (order) dynamics among neurons across layers, unfolding inside the entire network under the influence of (a) the input distribution, and (b) the intrinsic operation of the underlying computing substrate (e.g., AI accelerator).

Accounting for neuron execution scheduling strategies in-training is likely a missing link for between neural algorithms/models and neuromorphic hardware architecture co-optimization. This study merely scratched the surface of a research exploration in this direction, but it shows that unless we rethink our training methods to account for asynchronous dynamics and redesign neuromorphic accelerators to move away from layer synchronization primitives, SNNs and brain-inspired computing could fail to deliver both performance and efficiency.

It is worth noting that the discrepancy between the dynamics encoded in neurons' membranes and the way these dynamics are propagated across the neural network in current SNNs, was also independently identified and published only recently in (Yao et al., 2024; 2025). To address this perceived as "modeling flaw" in how SNNs work, they propose training ANN models and converting them to SNNs by translating quantized integer activations to spike bursts of equivalent cardinality; or directly training bespoke SNN models where neurons fire not single spikes but spike bursts proportional to their membrane potentials. While these approaches seem to at least preserve ANN performance, they are only suited for models of integrate-and-fire

(IF) neurons, since membrane leakage in LIF neurons would become ineffective with bursts of this sort. More importantly, the spike rates increase dramatically with adverse effect on energy consumption (which is a function of the spike density). By contrast, in the herein presented exploration we identify the "flaw" in the per-layer synchronization mechanism (which is a baggage from ANNs, but not inherently native to SNNs) that blocks these dynamics from expressing freely throughout the network. We show that casting away layer synchronization and enabling dynamics driven, adaptive neuron-execution scheduling during training *per-se*, we can train models aware of various asynchronous hardware, that finish inference faster than hardware agnostic ones, and provide correct predictions by processing only a subset of the spikes (improving both on energy and latency efficiency). Either approach however is in infancy, deserving more research and engineering for addressing scalability and demonstrating practical testing and efficiency with modern large deep models (e.g. as proposed in (Yao et al., 2025)) on neuromorphic accelerators that can accommodate them (e.g. (Davies et al., 2018; Furber et al., 2014; Tang et al., 2023)).

This motivates our follow-up work on challenges along 2 parallel directions: (a) improving on the computational resource limitations of asynchronous training with unlayered backpropagation, and (b) bridging this work to more complex contemporary model structures and constructs (e.g. (Deng et al., 2022; Yao et al., 2022; Guo et al., 2023b; Yu et al., 2022)). Regarding the former direction, recent advances with event-based back-propagation(Wunderlich & Pehle, 2021a) and sparse gradient learning (Lohoff et al., 2025), show promise for addressing these challenges.

## Acknowledgments

The herein research was partially sponsored and funded by Imec Netherlands, and the EU's Horizon Europe Research and Innovation programme (under Grant Agreement 101070679). Guangzhi Tang is partially funded by the Dutch Research Council's programme AiNed XS Europe programme (under Grant agreement NGF.1609.243.044, https://doi.org/10.61686/MYMVX53467). Roel Koopman is funded by the Dutch Research Council (under Grant agreement KICH1.ST04.22.021).

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

# A    Appendix / supplemental material

## A.1    Spiking neural networks

### A.1.1    Incoming current

For every neuron layer, all the synaptic weights on its inbound connections are kept in a weight matrix $\mathbf{W}^{(l)} \in \mathbb{R}^{N^{(l)} \times N^{(l-1)}}$, where $N^{(0)} =$ number of input features $N_{\text{in}}$. Using these weight matrices, the total incoming current $x$ for a neuron $i$ in the next layer can be computed using:

$$x_i^{(l+1)}[t] = \sum_{j=1}^{N^{(l)}} W_{ij}^{(l+1)} s_j^{(l)}[t] \tag{3}$$

where $s_j^{(l)}[t]$ is 1 if neuron $j$ in layer $l$ has emitted a spike at time $t$, otherwise 0.

### A.1.2    Membrane potential decay

The decay of the membrane potential is governed by a linear Ordinary Differential Equation (ODE). The analytical solution can be used to compute the decay:

$$u[t] = u[t - \Delta t] \cdot e^{-\frac{\Delta t}{\tau_m}} + x[t] \tag{4}$$

where $\tau_m$ is the membrane time constant and $\Delta$t the elapsed time.

### A.1.3    Threshold function

$$\Theta(u) = \begin{cases} 1 & \text{if } u > U_{\text{thr}} \\ 0 & \text{otherwise} \end{cases} \tag{5}$$

where $U_{\text{thr}}$ is the membrane potential threshold required for spiking.

## A.2    Event-driven state update rule for LIF neuron

When an input current is received at time $t$ by a neuron with a previous state $u[t_0]$ at some time $t_0 \leq t$, an atomic set of computations is executed. Start with computing the decayed membrane potential $u[t_-] = u[t_0] \cdot e^{\frac{-(t-t_0)}{\tau_m}}$, then update the membrane potential $u[t_+] = u[t_-] + x[t]$ with the input current $x[t]$, and finally set $u[t] = 0$ and emit a spike if $\Theta(u[t_+]) = 1$; otherwise $u[t] = u[t_+]$ without emitting a spike.

Note that the evaluation of the firing threshold takes place independently for every incoming synaptic current (instead of synchronizing and accumulating all currents first, and then evaluating the threshold). This allow to consider and preserve the temporal dynamics among the input currents (different arrival times) in the temporal derivative of the membrane potential of each neuron and propagate them accordingly.

### A.3 Arctan surrogate gradient

$$\Theta(u) = \frac{1}{\pi}\arctan(\pi u \frac{\alpha}{2}) \tag{6}$$

where $\alpha$ is a hyperparameter modifying the steepness of the function.

### A.4 Class prediction

Class prediction involves first calculating the output rates as follows:

$$c_i = \sum_{t \in T} s[i, t] \tag{7}$$

where $c_i$ is the spike count for class $i$, $T$ are all the timesteps for which a forward pass occurred, and $s[i, t]$ is the output of the neuron representing class $i$ in the output layer at the end of the forward pass at time $t$. For CIFAR-10, instead the value of $c_i$ is equal to the membrane potential of the output neuron for class $i$ at the end of having processed all the spikes. These values are subsequently used as logits within a softmax function:

$$p_i = \frac{e^{c_i}}{\sum_{j=1}^{N_C} e^{c_j}} \tag{8}$$

where $N_C$ is the total number of classes. The resulting probabilities are then used to compute the cross-entropy loss:

$$L = \sum_{i=1}^{N_C} y_i \log(p_i) \tag{9}$$

where $y \in \{0, 1\}^{N_C}$ is the target class in a one-hot encoded format.

## A.5 Parameters for the simulator

Table 4: Overview of all current simulator parameters. If the text is *italic*, then the parameter was not used for the experiments in this paper.

| Name | Value range | Description |
|---|---|---|
| Forward group size | $\mathbb{N}_{>0}$ | 3.1.2 |
| Scheduling policy | RS/MS | 3.1.2 |
| Prioritize input | True/False | If input spikes are propagated before any spikes from inside the network. |
| Stop condition | On spiking done / On output | 3.1.2 |
| Forward steps after output | $\mathbb{N}_{\geq 0}$ | 3.1.2 (only for "On output" stop condition) |
| Refractory dropout | $[0.0, 1.0]$ | 3.2.2 |
| Momentum noise | $\mathbb{R}_{\geq 0}$ | 3.1.2 (only for "MS" scheduling policy) |
| Membrane time constant | $\mathbb{R}_{>0}$ | 3.1 |
| Input spike dropout | $[0.0, 1.0]$ | 3.2.2 |
| *Network spike dropout* | $[0.0, 1.0]$ | The same as input spike dropout, but for spikes from inside the network, applied per forward step. |
| Membrane potential threshold | $\mathbb{R}_{>0}$ | 3.1 |
| Timestep size | $\mathbb{N}_{>0}$ | 3.1.1 |
| *Synchronization threshold* | $\mathbb{N}_{\geq 0}$ | Have neurons wait for a number of input currents (including barrier messages) before being allowed to fire. Can be set per neuron. |
| *Emit barrier messages* | True/False | Have neurons emit a barrier message if they have retrieved exactly the number of input currents to exceed the synchronization threshold, but not enough to exceed the membrane potential threshold. |

Table 5: Simulator parameters for simulating neuromorphic processors

| Synchronization type | Neuromorphic processors | Forward group size | Synchronization threshold [emit barrier messages] |
|---|---|---|---|
| Barrier-based / layer | Loihi (Davies et al., 2018), SENECA (Yousefzadeh et al., 2022), POETS (Shahsavari et al., 2021) | # neu/layer | # neu/layer [True] |
| Timer-based / core | SENECA (Yousefzadeh et al., 2022), SpiNNaker (Painkras et al., 2013), TrueNorth (Akopyan et al., 2015), POETS (Shahsavari et al., 2021) | # neu/core | # neu/core [True] |
| Asynchronous | Speck (Caccavella et al., 2023), µBrian (Stuijt et al., 2021) | # neu/layer | 1 [False] |

### A.5.1 Neuron execution scheduling policy & asynchronous processing dynamics

A scheduling policy in the simulation environment aims to capture the biases introduced by the operating principle of an AI accelerator (class) on data dynamics, and provide a means to reflect them in the model training process (see Unlayered Backpropagation). We experimented with two scheduling policies that capture

two different aspect of order dynamics, aiming to highlight their effects and importance but they are not the only possible. It is a topic of open exploration for the future, and co-design for dataflow AI accelerators.

- *Random Scheduling*, reflects the fact that the spike propagation and neuron integration process is or can be inherently noisy (either epistemically or aleatorically), thus affecting the temporal/rank order of spikes. It can be seen as annealing noise or dropout noise, which beyond a certain level breaks the system down but in small enough quantities (relevant to the vector pipelining of an accelerator – $F$-group size, arbitration between cores, network-on-chip routing, etc), makes the system more robust. Note that it is applied across the entire network, not within each layer.

- *Momentum Scheduling*, puts emphasis on the relative order dynamics in neuron evaluation as reflected in their membranes (i.e. which neuron is likely to fire first next – relevant to TTFS and rank-order codes for the importance of the first spike(s) even though we work with rate codes). Note again this takes place across all layers, and remains unbiased by artificial layer-synchronization barriers.

### A.6 Details on the experimental setup

#### A.6.1 Datasets and network architectures

The Neuromorphic MNIST (N-MNIST) dataset captures the MNIST digits using a Dynamic Vision Sensor (DVS) camera. It presents minimal temporal structure (Iyer et al., 2021). It consists of 60000 training samples and 10000 test samples. Each sample spans approximately 300 ms, divided into three 100 ms camera sweeps over the same digit. Only the initial 100 ms segment of each sample is used in this study.

The Spiking Heidelberg Digits (SHD) dataset (Cramer et al., 2020) is composed of auditory recordings with significant temporal structure. It consists of 8156 training samples and 2264 test samples. Each sample includes recordings of 20 spoken digits transformed into spike sequences using a cochlear model, capturing the rich dynamics of auditory processing. The 700 cochlear model output channels are downsampled to 350 channels.

The DVS gesture dataset (Amir et al., 2017) focuses on different hand and arm gestures recorded by a DVS camera. Like the SHD dataset, it has significant temporal structure. It focuses on 11 different hand and arm gestures recorded by a DVS camera. It consists of 1176 training samples and 288 test samples. The $128 \times 128$ input frame is downsampled to a $32 \times 32$ frame.

CIFAR-10 (Krizhevsky, 2012) is a widely-used image classification dataset consisting of 60000 32x32 color images across 10 classes, including animals and vehicles. It has no temporal structure. Unlike the other datasets, all input is provided in one single timestep.

For all four datasets, events of a data point belong to a continuous stream where they have a timestamp and an index position (see figure 8). In the case of N-MNIST, the index corresponds to a position within a $34 \times 34$ pixel frame, with each pixel having a binary polarity value (either 1 or 0), leading to a total of $34 \times 34 \times 2 = 2312$ distinct input indices. For SHD, the index denotes one of the 350 frequency channels. For DVS gesture, the $128 \times 128$ input frame is downsampled to a $32 \times 32$ frame. Like N-MNIST, each pixel has a binary polarity value, so in total this gives $32 \times 32 \times 2 = 2048$ input indices. Unlike the other datasets, for CIFAR-10, the input events present a continuous value (so they are currents instead of spikes), while also still being assigned an index and timestamp (although the timestamp is irrelevant). For encoding them as inputs to the models the stream of events is sliced in time-bins $(\epsilon_k, \epsilon_{k+1}, ...)$ along the temporal dimension and the timestamps are discretized to the time-bin index. The time-bins then construct time-frames whereby all the events at each spatial index are accumulated (counted). Each time-frame is then used as input to the model at discrete subsequent timesteps. This is one of the common-place input encoding for SNNs.

#### A.6.2 Performance metrics

To evaluate accuracy, output rates $c_i$ for each class $i$ are first calculated as outlined in equation 7. The predicted class corresponds to the one with the highest output rate. Accuracy is then quantified as the ratio of correctly predicted outputs to the total number of samples.

Table 6: Network architecture and hyperparameters. The architecture is given as [neurons in hidden layers × number of hidden layers] - [neurons in output layer].

|  | N-MNIST | SHD | DVS gesture | CIFAR-10 |
|---|---|---|---|---|
| Architecture | [64×3]-10 | [128×3]-20 | [128×3]-11 | see A.10 |
| Timestep size | 10 ms | 10 ms | 20 ms | N/A |
| Batch size | 256 | 32 | 32 | 64 |
| Epochs | 50 | 100 | 70 | 150 |
| Learning rate | 5e-4 | 7e-4 | 1e-4 | 2e-4 |
| Membrane threshold $U_{\text{thr}}$ | 0.3 | 0.3 | 0.3 | 0.2 |
| Weight decay constant $\lambda_W$ | 1e-5 | 1e-4 | 1e-5 | 0 |
| Membrane time constant $\tau_m$ | 1 ms | 100 ms | 100 ms | N/A |
| Surrogate steepness $\alpha$ | 2 | 10 | 10 | 2 |
| Input spike dropout | 0.25 | 0.2 | 0.2 | 0 |
| Forward group size $F$ | 8 | 8 | 8 | 512 |
| Refractory dropout | 0.8 | 0.8 | 0.8 | 0.7 |
| Momentum noise $\lambda_{\text{MS}}$ | 1e-6 | 0.1 | 0.1 | see A.10 |

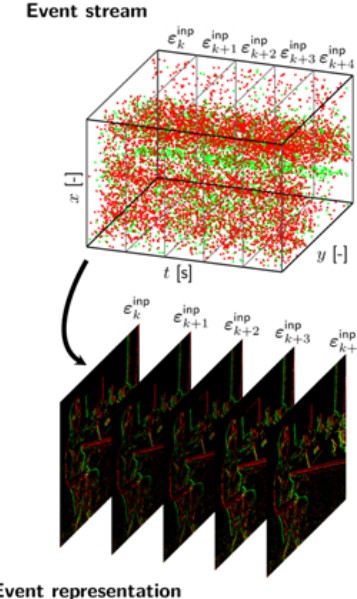

Figure 8: Encoding of a continuous temporal stream of input events into frames (Hagenaars et al., 2021).

Spike density is computed using:

$$\text{density} = \frac{1}{N_{\text{samples}} \cdot N_{\text{neurons}}} \sum_{i=1}^{N_{\text{samples}}} N_{\text{spikes}}[i] \tag{10}$$

where $N_{\text{samples}}$ is the number of samples, $N_{\text{neurons}}$ is the number of neurons in the hidden layers, and $N_{\text{spikes}}[i]$ is the total number of spikes during inference of the sample $i$.

## A.7 Results on hyperparameters

Choosing a smaller $F$ (i.e., with a more asynchronous system), may improve accuracy, particularly benefiting models with mechanisms that rely on network asynchrony such as momentum scheduling. However, reducing

$F$ also has its drawbacks. It significantly raises resource demands (discussed in section A.8), and there is a risk of reducing the effectiveness or even stalling the training process, as observed for SHD and DVS gesture.

Refractory dropout, can positively affect training outcomes. An explanation for this is that it increases the gradient flow by allowing more spiking activity. However, using full refractory dropout can also reduce performance, likely due to the inability to generalize to inference with refractoriness.

The momentum noise helps by introducing a slight stochastic element into the spike selection process, helping to avoid potential local minima that a purely deterministic selection method is prone to get stuck in. This seems to do little for N-MNIST, but for more complex datasets like SHD and DVS gesture it has a significant effect.

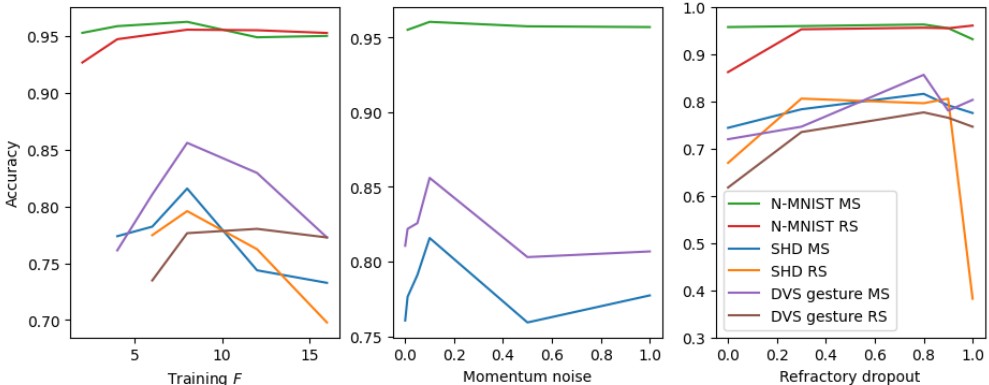

Figure 9: Results from inference with network asynchrony for different hyperparameters used during training. For SHD and DVS gesture with random scheduling, training $F$s smaller than 6 are not included due to the training failing to converge.

## A.8    Results on resource use

Table 7: Resource usage compared between layered and unlayered backpropagation during the second epoch of training on an NVIDIA Quadro RTX 5000.

| Method | Time per epoch (s) | VRAM use (MB) |
|---|---|---|
| **N-MNIST** | | |
| Layered | 14 | 210 |
| Unlayered RS, training $F = 16$ | 19 | 412 |
| Unlayered RS, training $F = 8$ | 24 | 556 |
| Unlayered RS, training $F = 4$ | 37 | 986 |
| **SHD** | | |
| Layered | 38 | 214 |
| Unlayered RS, training $F = 16$ | 118 | 2220 |
| Unlayered RS, training $F = 8$ | 292 | 4844 |
| Unlayered RS, training $F = 4$ | 681 | 10574 |
| **DVS gesture** | | |
| Layered | 120 | 244 |
| Unlayered RS, training $F = 16$ | 150 | 3802 |
| Unlayered RS, training $F = 8$ | 177 | 6984 |
| Unlayered RS, training $F = 4$ | 245 | 13914 |

The increase in processing time is less pronounced for the N-MNIST and DVS gesture datasets compared to the SHD dataset. This discrepancy could be due to computational optimizations that apply specifically to the N-MNIST and DVS gesture datasets (both being vision-based datasets).

### A.9 Results on error convergence

Both unlayered training with random scheduling and momentum scheduling achieve lower loss values compared to layered training, as shown in figure 10. At the start of training, random scheduling shows a slightly less steep loss curve, while momentum scheduling shows a steepness comparable to that of layered training. For CIFAR-10, error convergence is shown in figure 12.

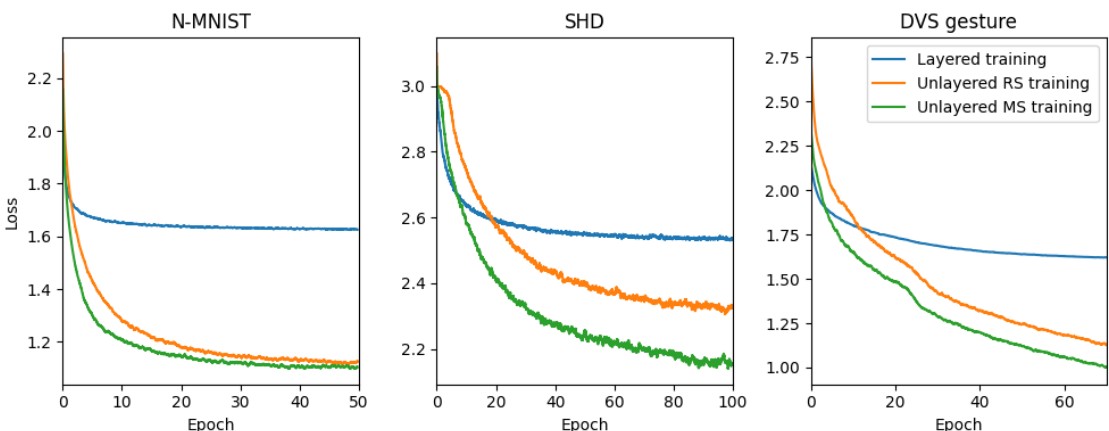

Figure 10: Error convergence for different synchronization methods.

### A.10 Details on CIFAR-10 dataset with VGG experiments

The details of network structure of the VGG model that we used to train on the CIFAR-10 dataset is shown in figure 11. We have removed (for simplicity) the batch normalization and dropout layers (that explicitly necessitate layer synchronization).

To reduce the memory overhead for training with unlayered backpropagation and make it feasible on an NVIDIA RTX 5000 we provided all the input of each sample in a single timestep (thus eliminating the unfolding across timesteps), and we adopted a forward group size $F = 512$ for training and $F = 64$ for testing. Additionally we deployed Integrate-and-Fire (IF) neurons that do not leak.

Output predictions were based on the membrane potential accumulated at the output neurons. Because of the very large $F$-group size, MS scheduling policy was difficult to train requiring rather large amounts of annealing noise (the range of values tested are shown in figure 12). Note that a large $F$-group is typically adverse for supporting network asynchrony, but in this experiment we were limited by the training resources given the currently high computational cost of unlayered backpropagation. Nonetheless, the results in figure 7, show that even with such large value range for $F$-group, there is a monotonic trend for accuracy recovery in asynchronous inference, as we decrease $F$.

The training error evolution is reported in figure 12. For momentum scheduling with $\lambda_{MS} = 0.1$ or $\lambda_{MS} = 2.5$, training was unstable after initial convergence and was terminated as soon as the error started to diverge. This was not the case for random scheduling or larger values of $\lambda_{MS}$.

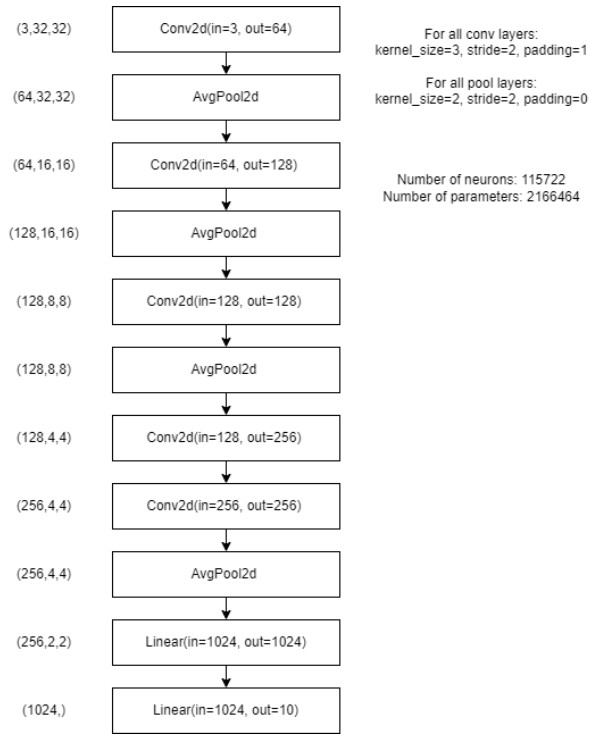

Figure 11: Simplified VGG network topology without batch normalization and dropout layers.

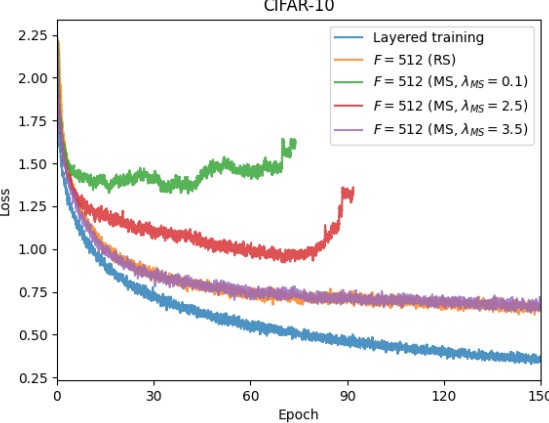

Figure 12: Training error convergence for CIFAR-10 for different synchronization methods and amounts of momentum noise in the MS scheduling policy.

### A.11 Neuromorphic processing and energy efficiency

Neuromorphic chips such as µBrain (Stuijt et al., 2021) and Speck (Caccavella et al., 2023) are at the forefront of asynchronous spiking neural network (SNN) hardware development, showcasing significant benefits over traditional clock-based architectures. Both chips employ an event-driven, fully asynchronous, architecture, eliminating the need for global or per-layer synchronization. In both cases, all neurons can fire a spike immediately once their membrane potential crosses a specific threshold in response to integrating an incoming current/spike (without waiting for a synchronization signal or an interval timer to expire). Such fully event-driven inference allows independent and instantaneous neuron processing, reducing computational

overhead and latency, and directly matches the operational principles of SNNs. However, neurons with fully event-driven asynchronous computation are sensitive to the sub-microsecond timing and exact order dynamics of the incoming spikes, which a time-stepped training algorithm is not sensitive to. As a result, there will be a significant accuracy drop if the deployed SNN of these chips is trained with a time-stepped algorithm.

Traditional synchronous simulation methods for SNNs introduce limitations when mapping to these asynchronous neuromorphic chips. In synchronous simulations, time is discretized into uniform steps, and all neurons in a layer are updated simultaneously, leading to excessive idle computations and artificial delays that are not representative of "real-time" temporal dynamics. In contrast, asynchronous simulation methods align more closely with the intrinsic event-driven nature of neuromorphic hardware. They allow all neurons to react as events occur, mirroring the operational principle of chips like μBrain and Speck. This results in lower latency and more efficient mapping of SNN models onto the hardware.

In the absence of common-place asynchronous training and simulation methods, many other neuromorphic chips (e.g. SpiNNaker (Furber et al., 2014), TrueNorth (Akopyan et al., 2015), Loihi (Davies et al., 2018), Seneca (Tang et al., 2023)) resort to explicit layer synchronization primitives (timer-based current integration or explicit signaling) for maintaining consistent performance with synchronous simulated and trained models. Under this constraint these systems execute models event-based, but not end-to-end asynchronously (even though they could). This comes at the cost of increased latency of execution and energy consumption.

Typically, energy consumption on such digital neuromorphic accelerators has two components. A baseline static one that is present even when the system is idle so long as it is powered, and which relates to its hardware attributes and components (memory leakage, clock frequency, manufacturing technology, and other). The second is a dynamic component that relates to the operation of the system for executing a model. The latter has to do with actual computations the system performs, memory I/O (typically the source of the Von Neummann bottleneck), and communication (particularly in multicore systems that entail a network-on-chip). In many architectures, a technique called power-gating helps save substantial energy from static power by switching off parts of the system that are idle.

Model execution is directly related to energy consumption due to dynamic power (e.g. when fetching data from memory and executing arithmetic operations), but also indirectly due to static power (as a function of the inference latency that forces the system to stay powered).

