# OpenReview forum: "Exploring the Limitations of Layer Synchronization in Spiking Neural Networks"
_TMLR — Accepted by TMLR_

### Review · Reviewer_d2oa · 2025-03-30

**Summary Of Contributions:**

This work presents and quantifies the Layer Synchronization problem in SNNs, then explores a potential solution direction, based on a generalization of backpropagation-based training that integrates knowledge about an asynchronous execution scheduling strategy for learning models suitable for asynchronous processing. Experimental results show that the proposed method can improve performance, inference speed, and energy efficiency.

**Audience:**

Yes

**Claims And Evidence:**

Yes

**Requested Changes:**

Please see weaknesses.

**Strengths And Weaknesses:**

Strengths: The work is clearly written, and the proposed method leads to several improvements in SNNs.

Weaknesses: my biggest concern with this work is that the problem the authors postulate actually already has better solutions. For example, synchronous integer training and asynchronous spike inference. That is, assuming that spiking neurons emit integer spikes during training, they can be equivalently converted into asynchronous spike trains during inference. Speck's software Sinabs supports this approach (page 13 of the paper [1]). In addition, recent work has analyzed the advantages of synchronous integer training and asynchronous spike inference, which can simultaneously bring improvements in five aspects [2]: training consumption, inference power consumption, scalability, performance, and support for asynchronous neuromorphic chips. More importantly, doing so can support the training of large-scale SNNs and help them perform very well [3,4]. In this work, the authors' experiments are still tiny. What do the authors think of this?

---
[1] Spike-based dynamic computing with asynchronous sensing-computing neuromorphic chip. In Nature Communications, 2024.

[2] Scaling Spike-driven Transformer with Efficient Spike Firing Approximation Training. In IEEE TPAMI, 2025.

[3] Integer-Valued Training and Spike-Driven Inference Spiking Neural Network for High-performance and Energy-efficient Object Detection. In ECCV 2024.

[4] Spike2Former: Efficient Spiking Transformer for High-performance Image Segmentation. In AAAI 2025.

---

> ### Author Response · Authors · 2025-04-15
> **Why not a solved problem and need for better solutions**
>
> We appreciate the insightful feedback and opportunity to clarify how our contributions are distinct from existing methods. We respectfully disagree with the assessment that this is a solved problem with better solutions. And we argue below that the cited literature by the reviewer actually emboldens this view, namely the problem is currently far from being generically addressed, and it deserves broader attention and research.
>
> As correctly highlighted in [1] and [2], (pointed by the reviewer) there is a discrepancy between the dynamics encoded in the neurons' membranes and the way these dynamics are communicated across the network. A key difference between these works and ours is that they attribute the discrepancy to an assumed fundamental flaw in the spiking neuron activation mechanism (unitary spike), and address it by increasing the spike activation rate proportionally to the membrane for each neuron, while in our exploration we identify the flaw in the per-layer synchronization mechanism (which is a baggage from ANNs, but not inherently native to SNNs) that blocks these dynamics from expressing freely throughout the network. We show that casting away layer-synchronisation and enabling dynamics driven adaptive neuron-execution scheduling makes networks finish inference faster providing predictions with processing only a subset of the activations (which agrees with neuroscience motivations for energy and latency efficiency).
>
> We already acknowledged in our paper that Speck (pointed to in [1] and [2] by the reviewer, and in [5] also cited in our paper) is one of the known digital neuromorphic accelerators to support end-to-end fully-asynchronous processing. Another other one being uBrain (in [6] also cited in our paper). However, both of them can only support small networks, and each of them appears to require custom non-generically applicable training practices so as not to degrade substantially performance, even for only a few layers (5 used in Speck in [1] pointed by the reviewer, and 4 in [5], and 3 in the case of uBrain in [6]). Specifically, in the referred literature, the proposed integer activation training [2], and multi-spike firing [1,5] result in increased firing rates, and may lead to unstable gradients that call for normalization and special regularization on activations (issues documented in more detail in [5]). These training approaches essentially aim to generate gradients proportional to the neuron membrane voltages, and in similar spirit, in uBrain [6] training was performed directly using gradients of quantized membrane voltages to match integer firing rates during ANN-to-SNN conversion. Again this approach also results in increased firing rates inside the network.
>
> Increasing the firing rates, increases unnecessarily energy consumption when all spikes need to be processed, as well as latency. Furthermore, normalization and skip/concat connections when used during in inference (e.g. commonplace in transformers) is a point of layer synchronization, that further increase latency by blocking end-to-end asynchrony. Also if normed re-scaling of rates across all layers is used to push down firing rates it risks latent layers (as depth increases) becoming completely quiescent, making these training practices unstable and not scalable. Finally, most of these tricks do not apply directly in recurrent or stateful networks (and training both for Speck [1,5] as well as uBrain [6] has only been showcased with feed-forward topologies and simple IF-like neurons, which eliminate even the self-recurrency).
>
> There are several other neuromorphic architectures that support globally-asynchronous locally-synchronous (GALS) vectorised processing, cited in our paper, that are capable of asynchronous event-driven processing, but are typically used with layer-synchronization primitives (just because a generic effective training method for asynchronous inference is not available). These are time-multiplexed architectures that are much more scalable for deeper networks, but with inevitably higher I/O overheads and thus costlier energy budget per spike. Critically for these architectures, a training approach that results in increased spike rates, has substantially higher energy and latency impact (than on Speck or uBrain). There is no known literature to us that has demonstrated (even in simulation) end-to-end asynchronous processing on them, while the size type of transformer networks discussed in [2,3,4] can only run on those platforms (not Speck or uBrain). In fact in [5], for spiking tiny-yolo only part of the network could be hosted in Speck.
>
> So in summary, discussing about network asynchrony for large model architectures, in ICs that they do not fit in, using training methods that have only been evaluated asynchronously on small models for specific chips, is clearly not suggesting that "the general problem is solved". It is a (good) starting point, calling for more experimentation, insights, and research.

---

> ### Author Response · Authors · 2025-04-15
> **Main differences between our work and the pointed literature**
>
> Our work aims at extending the scope of the existing literature that was referenced, from another (more general) viewpoint, and bringing interesting observations and insights (and also challenges) for algorithm-hardware co-design, many of which have not been yet reported in past literature. More specifically, on explicit differences between our proposed framework of enabling asynchronous processing and the one in [1-4]:
>
> ## 1. Comparison to Existing Training Solutions (Speck’s integer training and spike inference)
>
> Our work differs fundamentally in the scope and nature of the problem we are addressing:
>
> ### 1.a. Integer-Valued Training vs. Native Asynchronous Training:
> The integer-valued training referenced by the reviewer, while indeed powerful, primarily aims to approximate spike activity using integer activations, thereby enabling **conversion** to asynchronous spikes at inference. Our proposed approach, on the other hand, directly incorporates asynchronous neuron execution dynamics **during training itself**. This allows the model to adapt directly to neuron activation dynamics that occur naturally during asynchronous execution, rather than relying on approximations or conversions post-training.
>
> ### 1.b. Generalizability to Various Asynchronous Scheduling Strategies:
> The referenced methods typically impose a uniform integer training method, focusing mainly on approximating synchronous training setups for asynchronous inference. Our method explicitly generalizes the backpropagation training approach to include knowledge about asynchronous execution schedules (such as the Random Scheduling (RS) and Momentum Scheduling (MS)). By doing so, our approach adapts inherently and directly to asynchronous execution strategies, simulating support for a wider range of neuromorphic hardware implementations that can use different event propagation scheduling policies.
>
> ## 2. Addressing Scalability and Experimental Setup
> The reviewer’s concern regarding scalability and “tiny” experiments is valid and understood. However, the argument, holds both in our case (due to compute resource limitations), as well as in the pointed out literature (the actual examples evaluated asynchronously on Speck and for which accuracy vs efficiency is reported are not deeper than 5 layers and with standard benchmark datasets that can be solved with not very deep FC/CNNs).
>
> In our work, the experiments were intentionally chosen to clearly illustrate and quantify the underlying problem of layer synchronization. Larger-scale experiments, while highly relevant, are indeed computationally demanding and acknowledged explicitly in the manuscript (Section 4.6):
>
> *“Unfortunately, the computational cost of unlayered backpropagation, in the current framework and implementation, is rather high.”*
>
> However, we believe the results convincingly demonstrate the existence, significance, and potential solution direction for the layer synchronization problem. The hope is that these results can activate a community effort to explore better acceleration solutions for the training algorithm, extend/validate our observations on larger models, and leverage optimized simulation environments. In fact, we anticipate that addressing these computational challenges could lead to broader applicability and better alignment with practical neuromorphic hardware implementations.
>
> ## 3. Novelty and Impact Beyond Existing Work
> While acknowledging the strength of recent solutions, we believe our contribution remains meaningful due to the following unique aspects:
>
> ### 3.a. Direct integration of asynchronous dynamics during training:
> Our method specifically accounts for the timing and ordering dynamics of asynchronous neuron processing within training. This direct integration avoids potential performance gaps that may occur when inference conditions differ significantly from those assumed during training (as shown clearly in Section 4.3 of our manuscript).
>
> ### 3.b. Quantification of the “Layer-Synchronization” problem:
> Our study is the first comprehensive and quantitative analysis of the problem caused by training with layer synchronization and deploying in asynchronous hardware (generically, and in a context that applies also to GALS architectures of digital neuromorphic accelerators). By clearly quantifying accuracy degradation, energy consumption, and latency trade-offs, we set a baseline against which other methods—including integer spike approximations—can be meaningfully compared.
> Moreover, we believe that recent advances with event-based back-propagation [7] and autograd [8], hold promise for enabling substantial computational improvements in our vanilla training method.

---

> > ### Author Response · Authors · 2025-04-15
> > **Proposed ammendments to the current manuscript**
> >
> > Summary of changes to be made in response to reviewer:
> >
> > - Explicitly discuss and compare/clarify the differences between integer-based synchronous training for ANN-to-SNN conversion as depicted in [1,2], and our asynchronous dynamics-based training approach.
> > - Clearly acknowledge the scalability issue, its implications, and future directions to resolve it.
> > - Clarify our unique contributions more explicitly within the context of recent literature cited by the reviewer.
> >  We thank the reviewer for this insightful critique, which has significantly improved the clarity and contextualization of our work.

---

> > > ### Author Response · Authors · 2025-04-15
> > > **List of cited references**
> > >
> > > References:
> > >
> > > [1] Spike-based dynamic computing with asynchronous sensing-computing neuromorphic chip. In Nature Communications, 2024.
> > >
> > > [2] Scaling Spike-driven Transformer with Efficient Spike Firing Approximation Training. In IEEE TPAMI, 2025.
> > >
> > > [3] Integer-Valued Training and Spike-Driven Inference Spiking Neural Network for High-performance and Energy-efficient Object Detection. In ECCV 2024.
> > >
> > > [4] Spike2Former: Efficient Spiking Transformer for High-performance Image Segmentation. In AAAI 2025.
> > >
> > > [5] Low-power event-based face detection with asynchronous neuromorphic hardware. arXiv preprint arXiv:2312.14261, 2023. Now published at IJCNN 2024.
> > >
> > > [6] μBrain: An event-driven and fully synthesizable architecture for spiking neural networks. Frontiers in neuroscience, 15:664208, 2021.
> > >
> > > [7] Event-based backpropagation can compute exact gradients for spiking neural networks. Sci Rep 11, 12829 (2021).
> > >
> > > [8] A Truly Sparse and General Implementation of Gradient-Based Synaptic Plasticity. arXiv:2501.11407 [cs.NE]

---

> > > ### Comment · Reviewer_d2oa · 2025-06-18
> > > **Manuscript update**
> > >
> > > Could you please confirm whether the revised version of your manuscript has been updated in the system? I am still seeing the earlier version and would like to make sure whether the system has not refreshed or the updated file has not yet been uploaded. Thank you for your clarification.

---

> > > > ### Author Response · Authors · 2025-06-18
> > > > **Regarding revision of the manuscript.**
> > > >
> > > > We are currently working on providing a revised version of the manuscript that incorporates the changes for all 3 reviews.
> > > > As we received the 3 reviews far apart from each other (we received the 3rd review comments just last week) we only started applying the changes now (per the recommendation from the journal), as well as drafting the responses to the 3rd review.
> > > >
> > > > So we kindly ask to bear with us a couple of days, as we are on a bit hectic schedule. We try to finish them asap and we will send a notification to all three reviewers. (We may need to ask for one extra week of time from the Action Editor, if that would be possible, to allow time for further updates after that).
> > > >
> > > > We hope that is tolerable.

---

> > > > > ### Author Response · Authors · 2025-06-20
> > > > > **Updated manuscript**
> > > > >
> > > > > We now provided a revised version of the manuscript where (in green) we have tried to compile the changes for the review comments.

---

> > > > > > ### Comment · Reviewer_d2oa · 2025-07-02
> > > > > > **Official Comment**
> > > > > >
> > > > > > I thank the authors for their detailed clarifications, which have satisfactorily resolved my concerns. On the whole, I consider this work to be a meaningful contribution to the community.

---

> > > > > > ### Comment · Reviewer_d2oa · 2025-07-02
> > > > > > **Official Comment**
> > > > > >
> > > > > > I thank the authors for their detailed clarifications, which have satisfactorily resolved my concerns. On the whole, I consider this work to be a meaningful contribution to the community.

---

### Review · Reviewer_t4gC · 2025-04-15

**Summary Of Contributions:**

This paper presents an investigation into the impact of asynchronous processing in Spiking Neural Networks (SNNs). The authors set up a framework to simulate the behavior of SNNs in which neurons can emit spikes at any time, in an event-drive process that is independent on other conditions (such as waiting for all spikes to arrive at other neurons) that could delay the propagation. A two-stage backpropagation algorithm is used for training, which combines a conventional BPTT (backpropagation through time) approach, with a second unrolling that processes groups of emitted spikes. It is shown that, in such asynchronous SNNs, neurons tend to fire earlier and to emit fewer spikes than in conventional (layered) SNNs, provided the process is terminated when the first output is emitted. It is also highlight the importance of training the networks with an asynchronous process (instead of running inference on conventionally trained networks) to improve accuracy and lower spike density. Experiments are provided on 4 benchmarks (N-MNIST, SHD, DVS, CIFAR10), presenting spatial, temporal, or spatial-temporal features.

**Audience:**

Yes

**Broader Impact Concerns:**

Not addressed in the manuscript. No concerns on my end.

**Claims And Evidence:**

Yes

**Requested Changes:**

See previous section.

**Strengths And Weaknesses:**

**Strengths**

- The subject of this investigation is of clear interest to the audience of this journal
- The paper is well structured, clearly written, and easy to follow
- To the best of my knowledge, the presented algorithm is novel
- Results showing the role of training asynchronous SNN to improve performance across two axes (both accuracy and energy efficiency) are convincing and could stimulate further research in this direction

**Weaknesses**
- Compared to conventional SSN, asynchronous SSN show more spikes when the network activity fully drains, but fewer if measured at the time the first spike is emitted by the last layer (Figure 3). While this observation is relevant for various SSN tasks, other tasks require measuring spike rate (rate coding) or the behavior of groups of spike (population coding), so I wonder how fair the "on output" metrics is. Can the authors comment on this point? In particular, how does the number of spike (~ energy efficiency) compares, as more outputs are taken into account, asymptotically approaching the "in spiking done" state?
- Table 3 shows some energy estimates; it is my understanding that the energy numbers are derived by simply multiplying the number of synaptic operations (estimated using the asynchronous algorithm) by a fixed value of energy consumption per operation (extracted from a paper). If this is correct, it should be made more explicit that these numbers aren't on-chip measurements.
- The main limitation of this algorithm is the significant overhead in training time (as properly acknowledged by the authors), which hinders its application to relatively small networks or restricted training configuration. What is the impact on inference time, both in terms of simulation and potential on-chip deployment?

**Other comments**
- DVS acronym (as in "DVS sensor") is first mentioned in the text in section 3.1.1 but only defined later in the Appendix

---

> ### Author Response · Authors · 2025-05-01
> **validity of results for predictions based on first-spike, rate-coding, and population coding**
>
> ***Review remark: Compared to conventional SSN, asynchronous SSN show more spikes when the network activity fully drains, but fewer if measured at the time the first spike is emitted by the last layer (Figure 3). While this observation is relevant for various SSN tasks, other tasks require measuring spike rate (rate coding) or the behavior of groups of spike (population coding), so I wonder how fair the "on output" metrics is. Can the authors comment on this point? In particular, how does the number of spike (~ energy efficiency) compares, as more outputs are taken into account, asymptotically approaching the "in spiking done" state?***
>
> We are thankful for the review feedback and questions raised, which give us the opportunity to better position and clarify our work and contribution.
>
> Before delving into a more detailed explanation we would like to disclaim that we do not advocate or suggest that predictions/inference should be based on first-output-spike. This is not a constraint for the work and results we presented.
>
> Rather, we merely report the observation that with successful training for asynchronous inference, the activity that comes out of the model (output layer), within only 1-2 forward steps, is leading to the same prediction, as when a model has been trained with layer synchronization, executed synchronously during inference, and the prediction is concluded on evaluating all output activity from the network (which is typically consistent with rate coding). The observation holds true both when we allowed neurons in the network to enter refractoriness after firing a spike as well as when there is no refractoriness.
>
> This suggests that the critical information (for making predictions) propagates faster/first to the output (i.e. has stronger flow dynamics), if the model has been successfully trained for asynchronous inference. And therefore one may already “safely” terminate inference after the first (few) spikes emerge at the output.
>
> The graphs in figure 4 encode this intuition. The x-axis shows essentially time/latency and amount of additional processing. I.e. as the number of forward steps increases more spikes are scheduled for processing (also leading to more spikes at the output layer). For F=8, 10 forward steps means about 80 more spikes are considered. The models were trained and are operated in inference as rate-based models, and the dashed lines show the accuracy attained when predictions are evaluated based on all spike activity at the output layer until a model becomes quiescent (“on spiking done” regime). The colored lines show how accuracy evolves with the number of forward steps to eventually/asymptotically approach the dashed line. For models trained for asynchronous inference this (and even higher) accuracy is achieved within one additional forward step after the first spikes emerge at the output layer. Letting more spikes to emerge at the output (e.g. and evaluating predictions based on rate) remains statistically close to the dashed line not improving further the accuracy. For synchronous trained rate based models the convergence to the dashed line is much slower and more incoherent across tasks (likely depends on the nature of the task, and/or where in the input stimulus the “key” information is – for example for SHD other in literature have also reported that most of the information content appears early in the stimulus, while in a visual task like MNIST one expects that unless most of the stimulus is consumed by the network for some digits it is hard to make a conclusive decision)
>
> Again the tasks that we tested involved primarily rate coding and also single/few spike codings. We did not experiment with population coding, but we think it would be surprising to see a different behavior in that case. Namely, under free flow of activation stimulus without synchronization barriers the sub-populations representing the correct classes/predictions should get stimulated first.

---

> > ### Author Response · Authors · 2025-05-01
> > **reported estimates in table 3**
> >
> > ***Review remark: Table 3 shows some energy estimates; it is my understanding that the energy numbers are derived by simply multiplying the number of synaptic operations (estimated using the asynchronous algorithm) by a fixed value of energy consumption per operation (extracted from a paper). If this is correct, it should be made more explicit that these numbers aren't on-chip measurements.***
> >
> > That is correct, the numbers do not come from on-chip measurements. They are projections  based on unit cost per SOP, where the unit cost comes from the on-chip measurements in [1] for uBrain.
> >
> > The factual energy consumption measured end-to-end on-chip for a digital accelerator would be typically higher than reported in table 3, if it would include static power consumption (leakage power even when the IC is not doing anything). The energy cost due to static power consumption is typically a function of the time the chip is powered, and would therefore add even more favorably for unlayered training and asynchronous processing due to the reduction in inference latency. However it is hard to quantify generically and as it varies with type of hardware components and design choices, as well as technology node. And so for the objectiveness of comparison in table 3, we left it out (i.e. number of SOPs is entirely algo/model dependent).

---

> ### Author Response · Authors · 2025-05-01
> **impact on inference time, both in terms of simulation and potential on-chip deployment**
>
> ***Review remark: The main limitation of this algorithm is the significant overhead in training time (as properly acknowledged by the authors), which hinders its application to relatively small networks or restricted training configuration. What is the impact on inference time, both in terms of simulation and potential on-chip deployment?***
>
> There is no (negative) consequence at inference time for on-chip deployment on the real hardware. Or more correctly there is a positive impact that inference will typically conclude faster (based on the observations we report). Architecturally, we defend the need however for leveraging adaptive scheduling on time-multiplexed accelerators, beyond mere fixed round-robin schedulers (this is inherently supported on accelerators with analog/digital neuron arrays).
>
> For the simulation, the simulated time is also not affected, while the wall time will be because of the continuous context switching for additionally simulating the dynamic scheduler or arbiter of an accelerator. Whether this means that inference in simulation would take longer (by outweighing the fact that less spikes are processed), or not, it is hard to generically assert for different types of workloads and sizes of models.

---

> > ### Author Response · Authors · 2025-05-01
> > **Summary of changes to be made in response to reviewer and references (from the answers above)**
> >
> > - Explicitly state that numbers in Table 3 are not coming from on-chip deployment of the models, rather based on energy per SOP from the literature.
> >
> > - Clarify that the reported results and observations are not limited to single-output-spike models but related also to rate-based trained models.
> >
> > - Clarify that unlike training, there is no overheads to be encountered from executing the proposed models for inference on neuromorphic hardware. From an architecture point of view layer synchronization barriers would need to be obsoleted and support for adaptive scheduling (other than just simple Round-Robin) would be needed. On non time-multiplexed accelerators, not even that need exists.
> >
> > We thank the reviewer for this insightful critique, which has significantly improved the clarity and contextualization of our work.
> >
> > References:
> >
> > [1] μBrain: An event-driven and fully synthesizable architecture for spiking neural networks. Frontiers in neuroscience, 15:664208, 2021.

---

> > > ### Comment · Reviewer_t4gC · 2025-05-02
> > > **response**
> > >
> > > I thank the authors for the well-crafted replies to my comments. I have no further questions at this time.

---

> > > > ### Author Response · Authors · 2025-06-20
> > > > **Updated manuscript**
> > > >
> > > > We provided a revised version of the manuscript where (in red) we have tried to compile the changes for the review comments

---

### Review · Reviewer_CxyB · 2025-06-12

**Summary Of Contributions:**

The authors in this paper highlight the challenge of executing models asynchronously when they were originally trained with layer synchronization, showing that this mismatch leads to degraded performance or diminished energy and latency benefits. They further propose a solution that incorporates knowledge of asynchronous execution scheduling strategy, enabling the development of models optimized for asynchronous processing. The experiments conducted highlights that  event-based asynchronous  models can be an efficient alternative to conventional AI solutions.

**Audience:**

Yes

**Broader Impact Concerns:**

There are no strong concerns on the ethical implications of the work.

**Claims And Evidence:**

No

**Requested Changes:**

1) More details should be given around how asynchronous computation is achieved for the experiments. Fig. 2 is not entirely convincing. The authors should provide a more illustrative figure for explaining asynchronous communication. Preferably code should also be attached.

2) More details regarding the simulations should be provided.

3) Good to have: Additional experiments on convolutional architectures.

**Strengths And Weaknesses:**

Strengths:
The motivation of the paper is strong. The paper explores a critical aspect of spiking neural networks—asynchronous event-based execution—which is fundamental to their energy and power efficiency.

Weaknesses:
1) How asynchronous computation is achieved in the experimental setup is not clearly highlighted and explained.
2) The authors do not perform any on-chip implementation. Furthermore, the simulation framework used is also not well established. I believe the authors should explain the simulation framework used in more details.
3) Architectures used are fairly simplistic (feedforward layers). It would be good to see some experiments using convolutional layers to understand how information aggregation will happen in those layers in an asynchronous setting.

---

> ### Author Response · Authors · 2025-06-20
> **How asynchronous computation is achieved in the experimental setup**
>
> ***Review remark: How asynchronous computation is achieved in the experimental setup is not clearly highlighted and explained.***
>
> A description of how asynchronous inference is achieved in the experiments (and implemented in the simulation environment) is detailed in section 3.1, and its subsections. In summary the main processing principles that enable it are event-driven processing of course; the fact that neurons evaluate their threshold immediately upon receiving a synaptic input (spike) from each fan-in synapse independently (as opposed to aggregating all synaptic inputs and the evaluating the threshold once – where they can often cancel each other out); the relaxation of forcing synchronization of processing at every layer; and the adoption of dynamic scheduling of neuron execution across the entire network which continuously adapts (stochastically) to flow dynamics of stimulus/activations or neuron membrane states. The last last two “ingredients” allow neurons deep inside the network to fire before processing at early layers has finished (if the flow dynamics encourage it).
>
> To convey this more succinctly in the revision we added such a summary of these main principles at the introductory part of section 3.1 and will apply some additional modifications in the existing text, where we thought clarity can be improved in response to the expectations by the reviewer.
>
> Meanwhile, in appendix A.6 (of the originally submitted version) we provided an illustrative explanation of how typical backpropagation contrasts with unlayered backpropagation. Therein the forward pass in unlayered backpropagation corresponds to Algorithm 1 in the main text, and depicts how inference is facilitated in the simulation environment. We suspect (based on the review remarks) that probably the illustration is also essential for contextualising the key concepts in the main text. Therefore we also moved the content from previously A.6 to the main text at the end of section 3.2.1. We hope that this addresses the reviewer’s concern, or else please suggest aspects that may need further improvement/clarification.
>
> Practically, for each of the experiments the input stimulus to the networks is supplied across a number of timesteps (except for the last experiment with VGG where the task was purely visual and the stimulus static, and therefore fed in a single timestep). Within each timestep the event-driven processing from input to output is carried out *not* in layer synchonized order (where neurons from one layer finish processing, then neurons of the second layer, and so forth), but instead based on the (4) principles summarized above. When deploying autograd this results in different structure of the compute graph unfolding in every timestep. This is in contrast to typical layer-synchronous processing, where the structure of the compute graph is the same in all timesteps; and hence results in (fundamentally) different gradient flow, and learning a different model (more robust for asynchronous inference). – This is what figure 7 illustrated in the original version (which has now been moved to the main text as figure 3).

---

> > ### Author Response · Authors · 2025-06-20
> > **Not on-chip implementation, and explaination of the simulation framework**
> >
> > ***Review remark: The authors do not perform any on-chip implementation. Furthermore, the simulation framework used is also not well established. I believe the authors should explain the simulation framework used in more details.***
> >
> > As we study the issue in its more general form we indeed have not performed on-hardware implementation. The simulation framework instead was developed specifically for simulating the inference of various (digital) neuromorphic accelerators capable of event-based processing, with any built in them constructs to enforce layer synchronization removed (such as timers in the case of Spinnaker [1], hardware barriers in the case of Loihi [2], or the soft across cores tick-mechanism in the case of Seneca [3]).
> >
> > The behavior of the simulation for asynchronous inference is described in section 3.1 (and its subsections) and the dataflow algorithm in Algorithm 1. We rephrased parts of these sections in hope of enhancing clarity. We  further added a summary of the essential functional principles of the simulator for leveraging asynchronous inference (and training) at the start of section 3.1. We also brought from the appendix to the main text previous figure 7 from the original submission that illustrates in c the forward inference event-loop of the simulation environment.
> >
> > The key parameterization of the simulation environment pertaining to asynchronous processing behavior of the different accelerators that we have considered and how they map to the behavior of various neuromorphic accelerators appear in Tables 4 and 5 in the appendix. We are unsure if bringing these tables in the main text creates clutter or if they are essential for clarity and so we kindly ask for the reviewer’s affirmation in this respect.
> >
> > Finally, the code is provided with the paper (URL in supplementary material) on OpenReview for inspection by the reviewers’ and reproducibility of the experiments; and will be made publicly available at the end of the review.

---

> > > ### Author Response · Authors · 2025-06-20
> > > **Experiments using convolutional layers**
> > >
> > > ***Review remark: Architectures used are fairly simplistic (feedforward layers). It would be good to see some experiments using convolutional layers to understand how information aggregation will happen in those layers in an asynchronous setting.***
> > >
> > > ***Feed-forward but with and without recurrence:*** While the topologies we used in the experiments are lacking explicit lateral connectivity, SNNs with LIF neurons are however inherently recurrent due to neuron state (which is equivalent to explicit self-recurrent connections). The stimulus in the first 3 experiments is provided to the networks across a number of timesteps, which is in accordance with RNN processing (i.e. exploiting the statefulness of LIF neurons to make use of the self-recurrency). We do not anticipate a different behavior from adding explicit lateral recurrent connections (since unfolding them will also lead to a feed-foward structure with additional layers with shared weight matrices), but the added complexity can make the communication of the key messages even more difficult to explain and defend.
> > >
> > > ***Conv layers:*** In regard to network connectivity while the first 3 experiments are with fully connected layers, the last experiment with the deeper VGG network involves (5) convolutional layers. The topology is provided in figure 11 in the appendix, alongside more details about this experiment (which entailed a high computational cost due to depth and the inefficient implementation of unlayered backpropagation). To increase the clarity about the use of convolutions we made small modifications in the main text to explicitly state the use of convolutional topology in this experiment.

---

> > > > ### Author Response · Authors · 2025-06-20
> > > > **Figure 2 explained and code.**
> > > >
> > > > *** Review remark: More details should be given around how asynchronous computation is achieved for the experiments. Fig. 2 is not entirely convincing. The authors should provide a more illustrative figure for explaining asynchronous communication. Preferably code should also be attached. ***
> > > >
> > > > We apologise for the confusion caused by Figure 2. It is not meant to illustrate asynchronous processing/communication, but rather highlight one of the key aspects that is fundamentally different in layer synchronous and asynchronous processing.
> > > > Specifically, layer synchronization warrants that activations from all neurons in a layer are available simultaneously to any post-synaptic neuron in the next layer. Hence they are integrated together and the threshold of the post-synaptic neuron is evaluated only once. As a result the currents arriving from different synapses often cancel each other out (given opposite sign of the weights), resulting in fewer spikes and different firing pattern than when each synaptic current is processed independently and likely triggering a spike, as is the case in asynchronous processing. In the case of SNNs the volume of spikes is reduced and the activation flow dynamics are dulled. This is not an issue (as least not as grave) in ANNs because activations communicate relative magnitudes.
> > > >
> > > > We updated accordingly figure 2 as well as the caption to try and exemplify the different effect (on firing) when currents are averaged together (in presence of layer synchronized processing) versus when currents are processed independently from individual synapses.
> > > >
> > > > The code provided with the manuscript (URL in the supplementary pdf) implements the simulator with the independently per synapse integration (which is the correct one in asynchronous processing).
> > > >
> > > > This code will be also made publicly available to allow experimentation and support reproducibility of the experimental results.

---

> > > > > ### Author Response · Authors · 2025-06-20
> > > > > **Summary of changes to be made in response to reviewer and references (from the answers above)**
> > > > >
> > > > > - Include a summary (at section 3.1 in the main text) of the primitives that leverage asynchronous processing for our experiments in the developed simulation environment, and will apply modifications to improve the clarity of the existing text in that regard.
> > > > >
> > > > > - Improve the graphic and caption of figure 2 to disambiguate what it communicates.
> > > > >
> > > > > - Bring in the main text (from the appendix) illustration and explanations to provide more details about the simulation environment.
> > > > >
> > > > > - Apply modifications in the main text to unambiguously communicate that the experiments involved recurrent and feed-forward processing of input stimuli and with fully connected and also convolutional topologies.
> > > > >
> > > > > We are content with the reviewer’s acknowledgement of the importance of the topic of the paper, and the assessment about our motivation for this exploration. And we are thankful for the feedback that allowed us the opportunity to improve the clarify of our work.
> > > > >
> > > > > References:
> > > > >
> > > > > [1] S. B. Furber, F. Galluppi, S. Temple, and L. A. Plana. The spinnaker project. IEEE doi: 10.1109/JPROC.2014.2304638.
> > > > >
> > > > > [2] Mike Davies, et al. Loihi: A neuromorphic manycore processor with on-chip learning. Ieee Micro, 38(1):82–99, 2018
> > > > >
> > > > > [3] Guangzhi Tang et al. Frontiers in Neuroscience, 17, 2023. ISSN 1662-453X. doi: 10.3389/fnins.2023.1187252.

---

> > > > > > ### Author Response · Authors · 2025-06-20
> > > > > > **Updated manuscript**
> > > > > >
> > > > > > We provided a revised version of the manuscript where (in blue) we have tried to address the review comments

---

### Author Response · Authors · 2025-06-20
**revised version of the manuscript**

We have uploaded a revised version of the manuscript in accordance with the review feedback. The changes have been color coded to be identifiable. Specifically the color codes correspond to reviewers as follows:

  - in blue: Reviewer CxyB
  - in red: Reviewer t4gC
  - in green: Reviewer d2oa

Source code is available at the URL provided in the supplementary material pdf.

A kind request: If viable we would like to ask for a few additional days allowance to respond to any further comments by the reviewers, given the limited time left after compiling the revision and draft the last reviewer responses.

---

### Decision · Action_Editor_JA6m · 2025-08-06

**Recommendation:** Accept with minor revision

**Additional Comments:**

This is a solid paper and the authors have done well addressing various comments they received over the course of the review/rebuttal process. The authors need to submit a camera-ready version with all of the promised changes incorporated into the manuscript.

**Audience:**

Yes

**Audience Explanation:**

Improved training of efficient SNNs is an important direction for energy efficient AI, and it would be of interest to many TMLR readers interested in such matters.

**Claims And Evidence:**

Yes

**Claims Explanation:**

This paper investigates the discrepancy between the synchronous processing used in most artificial Spiking Neural Networks (SNNs) and the asynchronous nature of biological neural systems. While layer synchronization is a common practice, a truly asynchronous system, where neurons fire independently upon receiving input, could offer significant benefits in terms of latency and energy efficiency. However, a models trained with synchronous methods often perform poorly when executed asynchronously.

The authors quantify this performance drop and explore a new training method that integrates asynchronous execution scheduling into the backpropagation process. The provide experiments with two different asynchronous scheduling strategies, showing that models trained with their method can achieve significant reductions in spike count and inference time while maintaining or even improving accuracy.

The authors do not provide any on-chip demonstrations, but their simulations are sufficiently convincing that the reviewers were in consensus that the claims in the paper had been supported, particularly after the rebuttals.